# Histone modification dynamics at H3K27 are associated with altered transcription of *in planta* induced genes in *Magnaporthe oryzae*

**Wei Zhang**[ID][ʘ], **Jun Huang**[ID][ʘ], **David E. Cook**[ID]*

Kansas State University, Department of Plant Pathology, Manhattan, Kansas, United States of America

ʘ These authors contributed equally to this work.

* decook@ksu.edu

**Data Availability Statement:** Illumina sequence reads have been deposited at the National Center for Biotechnology Information, Short Reads Archive (BioProject accession No. PRJNA646251).

## Abstract

Transcriptional dynamic in response to environmental and developmental cues are fundamental to biology, yet many mechanistic aspects are poorly understood. One such example is fungal plant pathogens, which use secreted proteins and small molecules, termed effectors, to suppress host immunity and promote colonization. Effectors are highly expressed *in planta* but remain transcriptionally repressed *ex planta*, but our mechanistic understanding of these transcriptional dynamics remains limited. We tested the hypothesis that repressive histone modification at H3-Lys27 underlies transcriptional silencing *ex planta*, and that exchange for an active chemical modification contributes to transcription of *in planta* induced genes. Using genetics, chromatin immunoprecipitation and sequencing and RNA-sequencing, we determined that H3K27me3 provides significant local transcriptional repression. We detail how regions that lose H3K27me3 gain H3K27ac, and these changes are associated with increased transcription. Importantly, we observed that many *in planta* induced genes were marked by H3K27me3 during axenic growth, and detail how altered H3K27 modification influences transcription. ChIP-qPCR during *in planta* growth suggests that H3K27 modifications are generally stable, but can undergo dynamics at specific genomic locations. Our results support the hypothesis that dynamic histone modifications at H3K27 contributes to fungal genome regulation and specifically contributes to regulation of genes important during host infection.

## Author summary

Fungal pathogens of crops and humans pose annual threats to our food and health. There are many steps to the host infection process, during which fungal pathogens display unique growth, and use specific genes to cause disease. Despite this knowledge, many aspects of how pathogens regulate their genome to enact this process remain unknown. Here, we demonstrate how chemical modification of lysine residues on the histone H3, which helps organize and control DNA usage, play an important regulatory role in the model fungal pathogen causing rice blast disease. Our analysis shows a significant association between genes important for host infection and H3 lysine 27 methylation. We show

**Funding:** This work was supported in part by the United State Department of Agriculture-National Institute of Food and Agriculture (USDA-NIFA) (award no. 2018-67013-28492) (https://nifa.usda.gov/), and by the National Science Foundation Division of Molecular and Cellular Biosciences – Systems and Synthetic Biology (award no. 1936800) (https://www.nsf.gov/bio/mcb/about.jsp) to D.E.C. The funders had no role in study design, data collection and analysis, decision to publish, or preparation of the manuscript.

**Competing interests:** The authors have declared that no competing interests exist.

that by experimentally changing histone modifications, many fungal genes normally used during plant infection are turned on outside of the host. Furthermore, we detail how histone modifications can change naturally in the fungus during plant infection. These findings help broaden our knowledge of genome regulation for these pathogens, and advances the goal of a more comprehensive understanding of the infection process.

## Introduction

Epigenetic regulation of gene expression is common across many organisms including animals, plants and fungi, regulating diverse processes such as development, disease, and environmental response [1–4]. In eukaryotes, nuclear DNA is packaged around histone proteins to form nucleosomes [5]. Post-translational chemical modifications to histone tails provide epigenetic information contributing to genome regulation [6]. These modifications contribute to active, poised, or silenced transcription by affecting DNA-protein interactions, referred to as chromatin, which contributes to DNA accessibility and positioning in the nucleus [7,8]. Acetylation of H3 lys-27 (H3K27ac) is associated with transcriptional activation [9], whereas trimethylation at H3 lys-27 (H3K27me3) is linked with transcriptional silencing [10,11]. Histone modifications may function alone but they can also influence other histone modifications, termed cross-talk, to produce different chromatin states giving rise to a histone code regulating DNA functions [6,12]. For example, in *Drosophila* and mouse H3K27me3 and H3K27ac are mutually exclusive, and H3K27ac can antagonize H3K27me3 mediated gene silencing [13,14].

The H3K27me3 mark is catalyzed and maintained by Polycomb group proteins, termed Polycomb Repressive Complex 1 and 2 (PRC1 and PRC2). While PRC1 appears to be absent in fungi, PRC2, first identified in *Drosophila* as an important developmental regulator, is present in many eukaryotes [2,15–17]. In the model filamentous fungus *Neurospora crassa*, three core components of PRC2 (*NcSet7*, *NcSu(z)12* and *NcEed*) are required for normal H3K27me3 deposition, and 130 genes of total 774 H3K27me3-marked genes are upregulated ~7.5-fold in the absence of H3K27me3 [18]. The genome of *Fusarium graminearum*, the causal agent of wheat head scab, codes numerous secondary metabolite gene clusters that are enriched for H3K27me3 modification, and deletion of the methyltransferase responsible for H3K27me3, *FgKmt6*, resulted in transcriptional activation of many clusters [10]. Similarly, reduction of H3K27me3 in the rice pathogen *F. fujikuroi*, led to the induction of several secondary metabolite gene clusters and formation of novel chemical compounds [19,20]. In addition to pathogenic interactions, H3K27me3 helps regulate gene expression in the fungus *Epichloë festucae* during endophytic growth in its host [21]. Given these observations and the described function of H3K27me3 in facultative heterochromatin, genomic regions that switch between repressed and active transcription [10,13,14,22], dynamic regulation of H3K27 modifications provides an attractive regulatory model.

*Magnaporthe oryzae* (synonym of *Pyricularia oryzae*) is a filamentous fungal plant pathogen, best known for causing rice (*Oryza sativa*) and wheat (*Triticum aestivum*) blast diseases that threaten world food security [23–27]. The initial stages of *M. oryzae* infection, including spore germination, appressorium development and leaf penetration are well understood, but many details of growth and development inside the plant remain unknown [28–31]. A central paradigm of plant-microbe interaction is the use of secreted proteins and chemicals by an invading organism to suppress host immunity, blunt the immune response and change host physiology to facilitate infection [32,33]. Numerous effectors identified in *M. oryzae* are highly expressed *in planta* but remain silent or lowly expressed *in vitro* (i.e. *ex planta*), a common

phenomenon across filamentous pathogens [34–38]. Little is known about the molecular mechanisms controlling repression of fungal effector when growing outside of the host, and their subsequent induction during host infection. A previous study in a wheat-infecting strain of *M. oryzae* showed H3K4me2/me3 was positively correlated with active gene expression *in vitro* [39]. Given that H3K4me2/me3 is a conserved modification marking active transcription, it provides little information addressing the mechanism of gene *in vitro* silencing and *in planta* induction. Whereas H3K27me3 contributes to secondary metabolite gene repression in fungi, how histone modification dynamics at H3K27 contributes to transcriptional regulation of effectors is largely unknown.

To address this knowledge gap, we tested two related hypotheses: that H3K27me3 is involved in suppressing effectors and other *in planta* induced genes during *in vitro* growth, and dynamic change between trimethylation and acetylation at H3K27 underlies the switch between a repressed and activated state (Fig 1A). Using a combination of fungal molecular genetics, epigenomics, and transcriptomics analysis, we demonstrate that histone modification dynamics at H3K27 can regulate specific genomic loci, often important during host infection, supporting the hypothesis.

## Results

### Trimethylation and acetylation at H3K27 occupy distinct genomic regions in *M. oryzae*

To understand the regulatory role of H3K27 modification, we performed ChIP-seq for H3K27me3 and H3K27ac to establish the genomic distribution of these marks (S1 Table). The genome-wide domains for H3K27me3 and H3K27ac occupied discrete genomic regions in *M. oryzae*, with H3K27me3 forming longer discrete blocks, and H3K27ac having a more continuous distribution but shorter domains (Fig 1B and 1C). A total of 1,381 H3K27me3 domains were identified, ranging in size from 0.3 to 115.6 kb (average 6.4 kb). Collectively they occupy 8.8 Mb (20.6%) of the 42.8 Mb *M. oryzae* Guy11 wild type genome (Fig 1c and 1D and S1A Fig and S2 Table). This H3K27me3 genome distribution is greater than that reported in *Drosophila* (~3–7.5%), *Arabidopsis* (~5.7%), and the saprophytic fungus *Neurospora* (~6.8%) [18,40,41], but lower than the plant pathogenic fungus *F. graminearum* (~33%) [10]. The H3K27ac domains, total of 9,512, were significantly shorter than those of H3K27me3 and cover 10.9 Mb (25.6%) of the *M. oryzae* genome (Fig 1C and 1D and S1A Fig and S3 Table). In addition, the average H3K27me3 ChIP signal was significantly stronger than that for H3K27ac (S1B Fig).

In relation to the functional genome, H3K27me3 domains predominantly marked transposable elements (TEs) with 51.5% of the annotated TEs being occupied by a H3K27me3 domain, contrasted by only 1.8% being marked by H3K27ac (Fig 1D). Within the TEs, the H3K27me3-ChIP signals were significantly stronger for LINE and LTR retrotransposons compared to DNA Class II elements, while H3K27ac signals showed no significant difference between the classes (S2A and S2B Fig). We found the H3K27ac domains almost exclusively associated with coding regions, marking promoters, terminators and 21.1% of annotated genes, while H3K27me3 was present at only 16.6% of genes (Fig 1D). There was no Gene Ontology (GO) enrichment for the H3K27me3 marked genes, but Kyoto Encyclopedia of Genes and Genomes (KEGG) analysis suggested that many H3K27me3 marked genes were involved with primary and secondary metabolism (S4 and S5 Tables). Profile plots of ensembles genes and TEs revealed that H3K27me3-marked genes and TEs had lower H3K27ac signals, and those lacking H3K27me3 displayed higher H3K27ac signal (Fig 1E and 1F). In

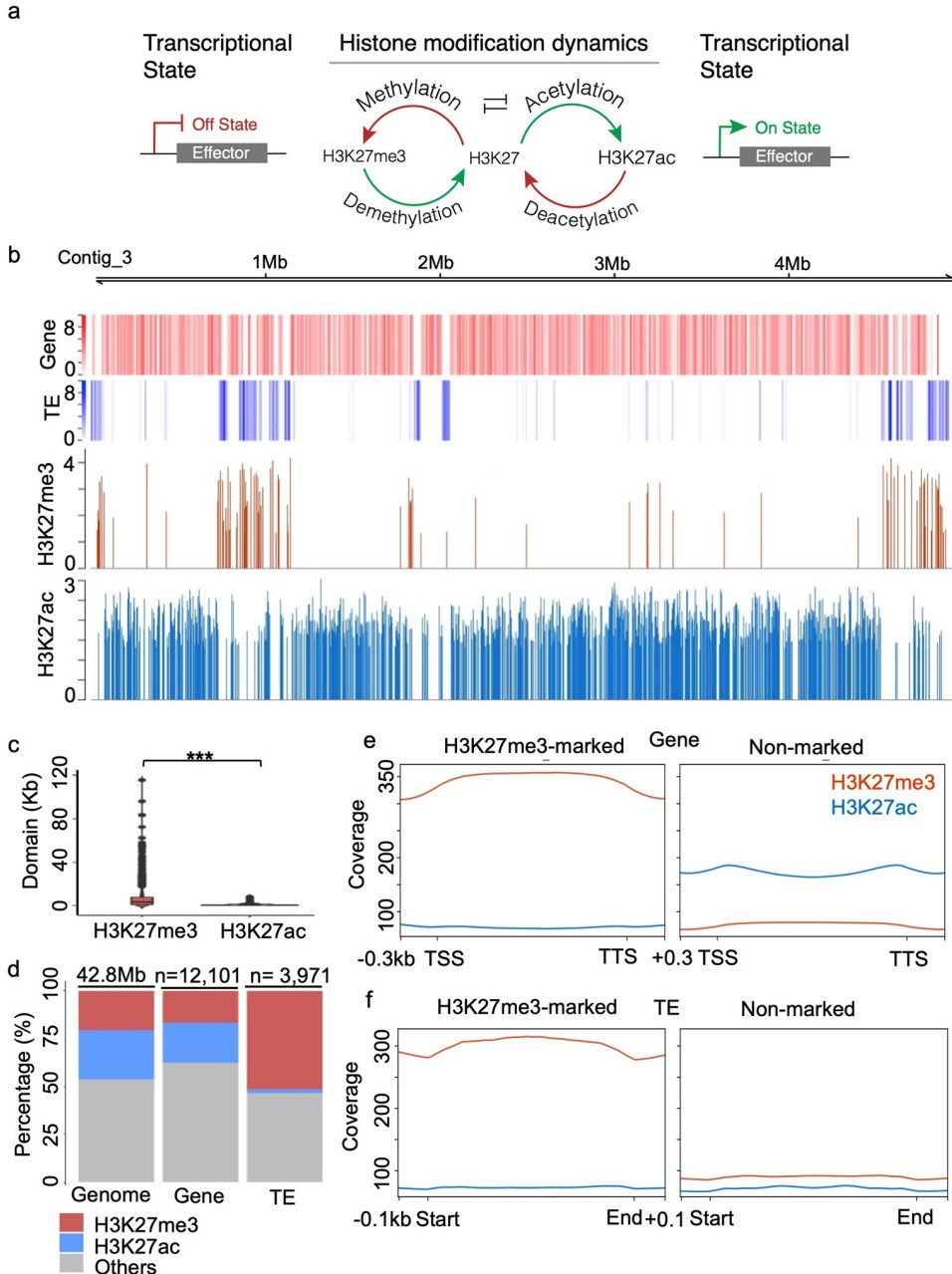

**Fig 1. Methylation and acetylation at H3K27 occupy repressive and active genomic regions with distinct distribution profiles.** (A) A working model for the dynamics of histone modification at H3K27 regulating fungal effector gene expression. (B) ChIP signals for H3K27me3 and H3K27ac occupy distinct genomic regions in *M. oryzae* Guy11 wild type growing in complete medium. (C) Violin plots illustrating genome-wide distributions of domains of H3K27me3 and H3K27ac. (D) Bar chart illustrating the percentage of genome (Geno), gene and transposable element (TE) marked by H3K27me3 and H3K27ac in wild type grown in complete medium. (E) and (F) ChIP signal profiles of H3K27me3 and H3K27ac on genes (E) and TEs (F) that marked (left) and non-marked (right) by H3K27me3.

addition, no genes or TEs were identified with both marks at H3K27 based on identified peaks, further showing the consistent and exclusive nature of the two marks. These results support the hypothesis that H3K27me3 is a repressive histone mark in *M. oryzae* and provides a distinct regulatory role from H3K27ac (Fig 1A).

## PRC2-mediated H3K27me3 contributes to normal *in vitro* growth

The core components of the PRC2 complex were identified in *M. oryzae* based on homology, termed *MoKmt6*, *MoSuz12*, and *MoEed*, and deletion mutants were generated for each (S3 and S4 Figs and S6 Table). Consistent with canonical PRC2 function, each of the three proteins were required for H3K27me3 (Fig 2A). Furthermore, the methyltransferase activity of PRC2 was

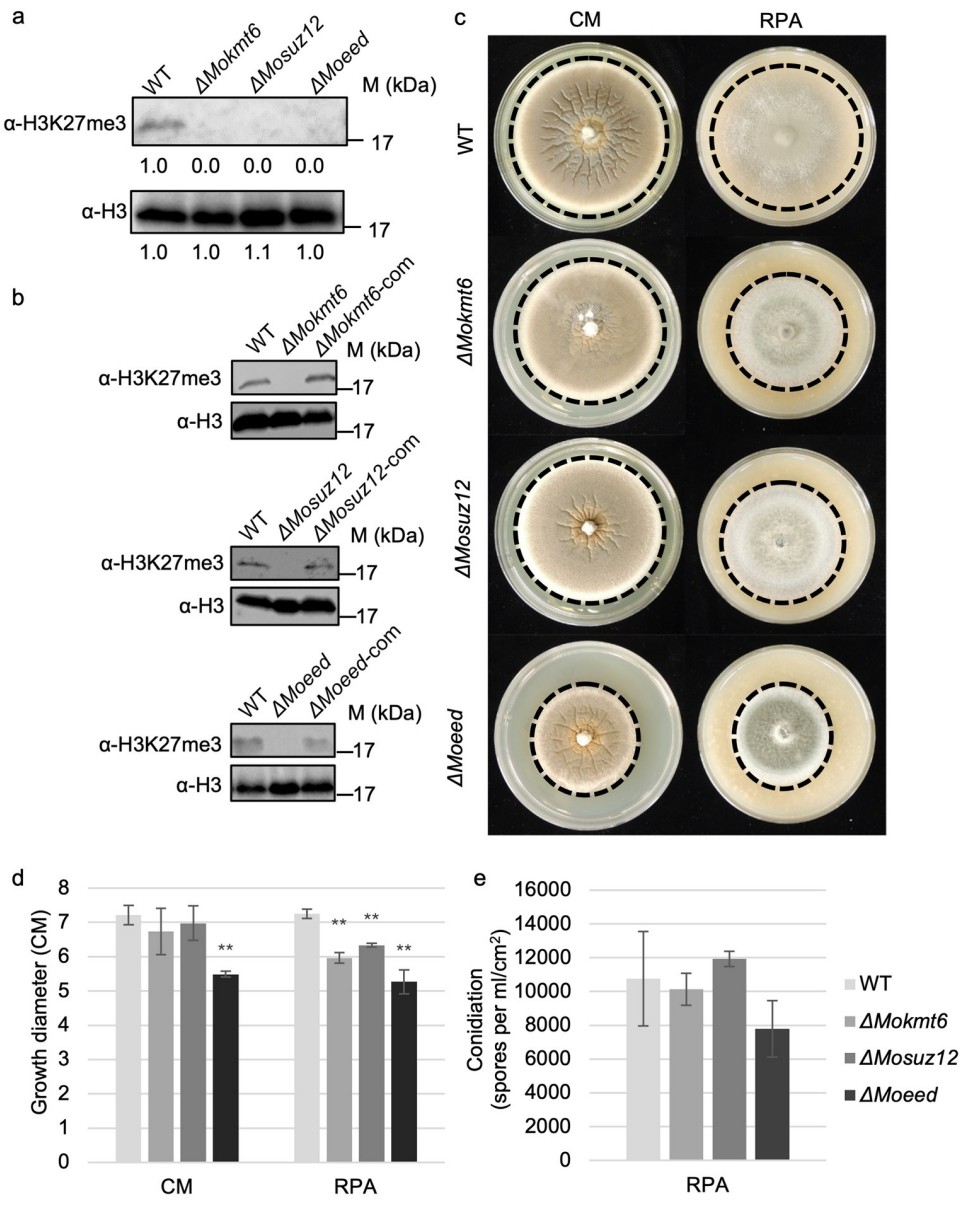

**Fig 2. PRC2 mediated H3K27me3 is required for normal fungal *in vitro* growth.** (A) Total proteins extracted from wild type (WT), *ΔMokmt6*, *ΔMosuz12* and *ΔMoeed* were used for western blotting. The anti-H3K27me3 was used for detecting the global level of H3K27me3 in the samples and anti-H3 was used as loading control. Intensity of signals were analyzed by ImageJ. (B) Total proteins extracted from wild type, *ΔMokmt6*, *ΔMosuz12*, *ΔMoeed*, *ΔMokmt6-com*, *ΔMosuz12-com* and *ΔMoeed-com* were subjected to western blotting. (C) Colony morphology of wild type, *ΔMokmt6*, *ΔMosuz12* and *ΔMoeed* on the complete medium agar (CM) and rice polish agar (RPA) at 12 days post inoculation. Dotted lines highlighted the edge of colonies. (D) Colony diameters were measured at 12 days post inoculation and subjected to statistical analysis. Significance is present by double asterisks ($p < 0.01$) or single asterisk ($p < 0.05$) to indicate significant differences compared with wild type by student's t-test. (E) Production of asexual conidia in different strains were measured after culturing on rice polish agar at 10 days post inoculation under constant light.

restored when the individual gene deletions were complemented (*ΔMokmt6*-com, *ΔMosuz12*-com *and ΔMoeed*-com) (Fig 2B and S5 Fig). Among the three mutants, only *ΔMoeed* showed significantly reduced growth on complete medium agar, while all three deletion mutants showed significant reduced growth on rice polish agar compared to the wild type (Fig 2C and 2D). The different growth rates observed on complete medium agar versus rice polish agar shows an interplay between PRC2 function and growth environment. Production of asexual conidia were not statistically significantly different for any mutant compared to wild type (Fig 2E).

## Loss of H3K27me3 causes redistribution of H3K27ac and H3K36me3 at specific genomic regions

In other eukaryotes, H3K27me3 and H3K27ac have antagonistic effects on transcription, and their dynamics provide an important transcriptional regulatory layer [13,14]. We observed an increase in H3K27ac signal in the three *M. oryzae* PRC2 mutants, which indicated a direct antagonism between the two modifications (Fig 3A). To determine if the altered H3K27ac levels were the result of global or local H3K27ac redistribution, ChIP-seq was performed on *ΔMokmt6*. Genome-wide Pearson correlation analysis showed H3K27me3 was negatively correlated with H3K27ac (r = -0.61, *p* < 2.2e-16, S6 Fig) in wild type, and the loss of H3K27me3 in *ΔMokmt6* caused a significant change in H3K27ac distribution (r = 0.73, *p* < 2.2e-16, S6 Fig). This further supports the antagonistic nature of the two modifications. We observed that the largest change in H3K27ac in *ΔMokmt6* occurred at genes and TEs that had previously been marked by H3K27me3 in wild type, in which these elements gained H3K27ac (Fig 3B). To monitor histone modification crosstalk, we checked the levels of another histone modification, H3K36me3, as it has been shown that H3K36me3 rarely co-exists with H3K27me3 [42–44]. We observed that H3K36me3 was positively correlated with H3K27me3 (r = 0.61, *p* < 2.2e-16, S6 Fig). The largest change for H3K36me3 in *ΔMokmt6* also occurred at genes and TEs that had been marked by H3K27me3 in wild type, but in this case these genes and TEs lost H3K36me3 (Fig 3B). Unsupervised K-mean clustering of genes based on H3K27me3 ChIP-seq from wild type resulted in cluster 1 and cluster 2, defined by the presence and absence of H3K27me3 in wild type respectively (Fig 3C). The wild type cluster 1 genes, containing the highest levels of H3K27me3, were devoid of H3K27ac, while genes in cluster 2 showed increasing H3K27ac as H3K27me3 decreased (Fig 3C and 3D). Interestingly, the genes marked by H3K27me3 in cluster 1 showed a significant gain of H3K27ac in *ΔMokmt6*, which showed the change from trimethylated to acetylated H3K27 occurred at specific loci (Fig 3C and 3D). The opposite was seen for H3K36me3 levels, which were highest in cluster 1, but decreased in the *ΔMokmt6* strain lacking H3K27me3 (Fig 3C and 3E). Similar overall patterns were observed for TEs, where there was a gain of H3K27ac in *ΔMokmt6* for TEs that had been marked by H3K27me3 in the wild type (S7 Fig). For example, TE rich regions on contig 5 contain H3K27me3 domains lacking H3K27ac in the wild type strain. However, these regions clearly gain H3K27ac and show a modest reduction for H3K36me3 when ChIP is conducted in *ΔMokmt6* (Fig 3F). Collectively, these results demonstrate specific regulatory machinery and rules are present governing H3K27 trimethylation and acetylation and crosstalk existing with H3K36me3. We did not observe a global redistribution of H3K27ac in the absence of H3K27me3, but rather local dynamics where previously trimethylated residues became acetylated.

## Differential gene expression upon loss of H3K27me3 and epigenetic state change

Given our empirical evidence that H3K27me3 and H3K27ac are exclusive, and that previously H3K27me3 marked genes can become H3K27ac in *ΔMokmt6*, we next assessed how this

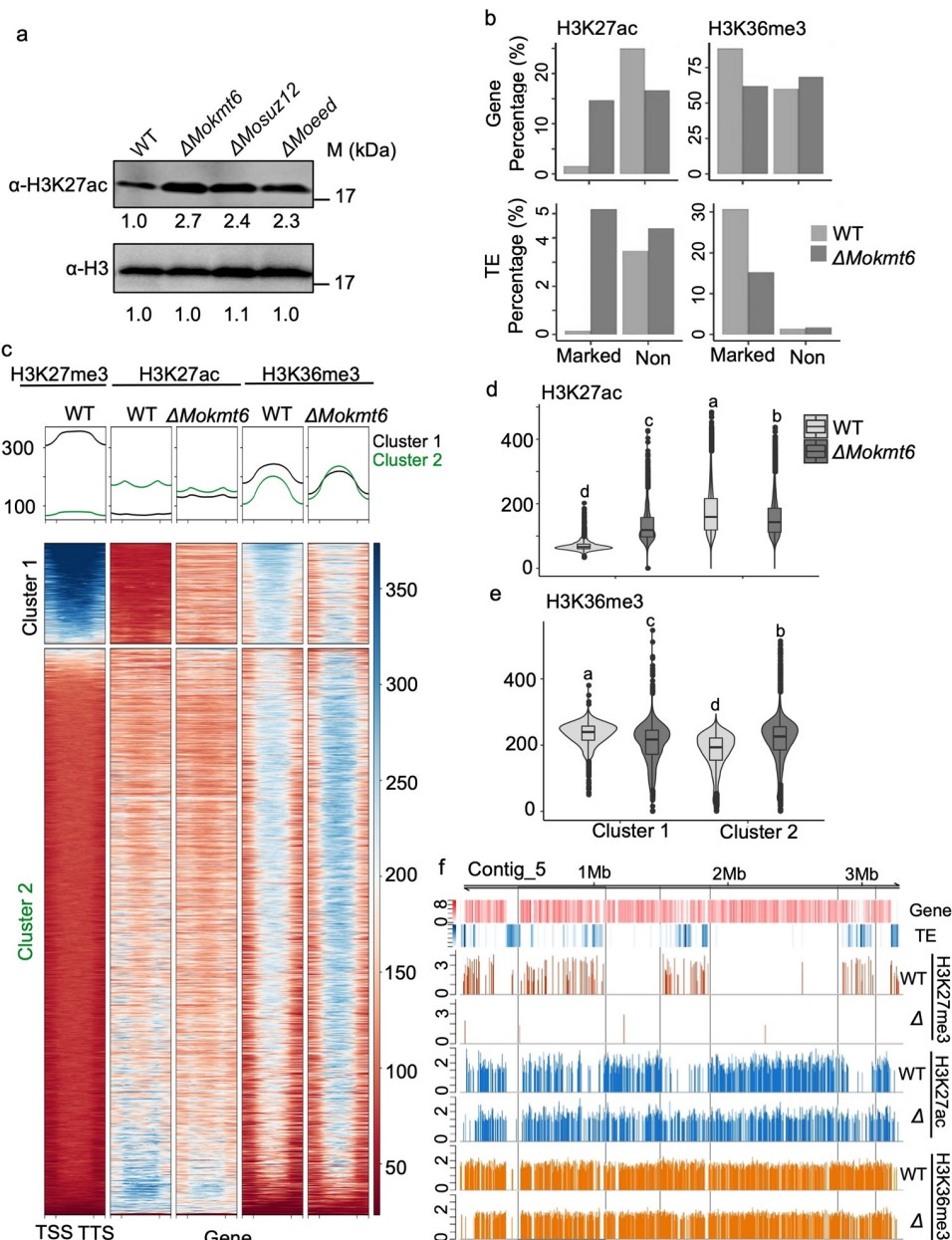

**Fig 3. Loss of H3K27me3 leads to redistribution of H3K27ac and H3K36me3.** (A) Western blotting analysis showed the global increased levels of H3K27ac in *ΔMokmt6*, *ΔMosuz12 and ΔMoeed* with absence of H3K27me3 compared with Guy11 wild type (WT). (B) Bar chart illustrating dynamics of H3K27ac (left) and H3K36me3 (right) marked genes and transposable elements (TEs) compared between wild type (blue) and *ΔMokmt6* mutant (red). (C) Heat maps visualizing profiles of ChIP signals for H3K27me3, H3K27ac and H3K36me3 across 12,101 genes in Guy11 wild type and mutant *ΔMokmt6*. Two clusters of genes are generated by unsupervised k-means analysis and ranked according to their H3K27me3 enrichment from wild type growing in liquid complete medium. (D) and (E) Violin plots illustrating redistribution of ChIP signals for H3K27ac and H3K36me3 in two clusters from the heatmap (C). Letters at the top indicating the significance. (F) ChIP signal showing the loss of H3K27me3 in mutant *ΔMokmt6* (*Δ*) resulted in the redistribution of H3K27ac and H3K36me3 at specific loci. ChIP signals are compared between Guy11 wild type and mutant *ΔMokmt6* growing in liquid complete medium.

change impacted transcription using RNA-seq. Gene expression analysis between wild type and *ΔMokmt6* grown *in vitro* identified that 1,115 genes were upregulated in the mutant (9.8% of transcriptome). Of these 1,115 up regulated genes, 980 (87.9%) of them were marked by H3K27me3 domains in the wild type, whereas only 9 in 269 downregulated genes in *ΔMokmt6* are marked by H3K27me3 (Fig 4A and S8 Fig and S4 Table). This shows significant local transcriptional repression by H3K27me3 in *M. oryzae*. We identified 229 genes that gained H3K27ac in the *ΔMokmt6*. This set of genes had significantly higher expression in *ΔMokmt6* compared to wild type (S9A and S9B Fig). Conversely, we identified 857 genes that lost H3K27ac in *ΔMokmt6*, which this set of genes was not significantly differently expressed between *ΔMokmt6* and wild type (S9C and S9D Fig). This suggested that gaining H3K27ac was associated with increased transcription, but H3K27ac is not required *per-se* for transcription as sites that lost the mark were not expressed at a lower level. We also found that 321 expressed genes lost H3K36me3 in *ΔMokmt6*, and this group of genes was significantly more highly expressed compared to wild type (S10 Fig). If we further characterize genes as having either only gained H3K27ac, only lost H3K36me3, or both, we see the same impacts on transcription (S11 Fig). That is, genes that gained H3K27ac or lost H3K36me3 were more highly expressed in *ΔMokmt6* compared to wild type under the same axenic growth conditions. These associations do not prove causative, but they are consistent with the hypothesis that histone modification dynamics contribute the transcriptional regulation.

To further characterize how the redistribution of H3K27ac to previously H3K27me3 sites impacted transcription, we identified and deleted the coding sequence for *MoGcn5*, the protein responsible for acetylating histone lysines, including H3K27ac (S12A and S12B Fig) [45]. The double deletion mutant, *ΔMokmt6ΔMogcn5*, lacks H3K27me3 and the majority of H3K27ac (Fig 4B). The double mutant also had an altered morphology and significantly reduced growth on complete medium compared to the wild type (S13A and S13B Fig). We also noted the double mutant has an elongated and flattened conidia morphology, while both the single mutants, *ΔMokmt6* and *ΔMogcn5*, have conidia that resemble the wild type (S13C Fig). Rice infection showed that both single mutants, *ΔMokmt6* and *ΔMogcn5*, produced smaller lesions on rice, while *ΔMokmt6ΔMogcn5* completely lost the ability to infect (S14 Fig). The double mutant's abnormal conidia and reduced growth likely resulted in the inability to infect rice. RNA-seq analysis of *ΔMokmt6ΔMogcn5* showed 29.5% of the H3K27me3 marked genes that were up regulated in *ΔMokmt6* were down regulated in *ΔMokmt6ΔMogcn5* (Fig 4A). These results were confirmed globally, where the average expression of H3K27me3 marked genes is significantly lower in wild type compared to *ΔMokmt6*, while the non-marked genes have a similar expression profile (Fig 4C). In *ΔMokmt6ΔMogcn5*, the average expression of H3K27me3 marked genes is significantly lower than in *ΔMokmt6*, showing the requirement for Gcn5 acetylation for part of the gene activation seen in *ΔMokmt6*. There is also a statistically significantly lower average expression for non-H3K27me3 marked genes in *ΔMokmt6ΔMogcn5* compared to wild type or *ΔMokmt6*, likely because Gcn5 acetylates additional lysine residues besides H3K27 that contribute to transcription [46,47].

To further investigate how H3K27 modification status links with individual gene expression, we integrated RNA-seq from the wild type, *ΔMokmt6*, and *ΔMokmt6ΔMogcn5* with the ChIP-seq (Fig 4D). Genes were sorted in descending order based on wild type RNA-seq levels, combined into groups composed of 100 genes each, and the values for RNA-seq and ChIP-seq were summarized. A clear positive association between H3K27ac and gene expression can be observed, with genes in the highest expression groupings having the highest H3K27ac signal and decreasing as transcription decreases ($r_{Pearson}$ = 0.67, $p < 2.2e-16$, Fig 4D). For H3K27me3, there is a non-linear relationship, with the line connecting each group average H3K27me3 signal remaining relatively flat until after approximately group 80, after which

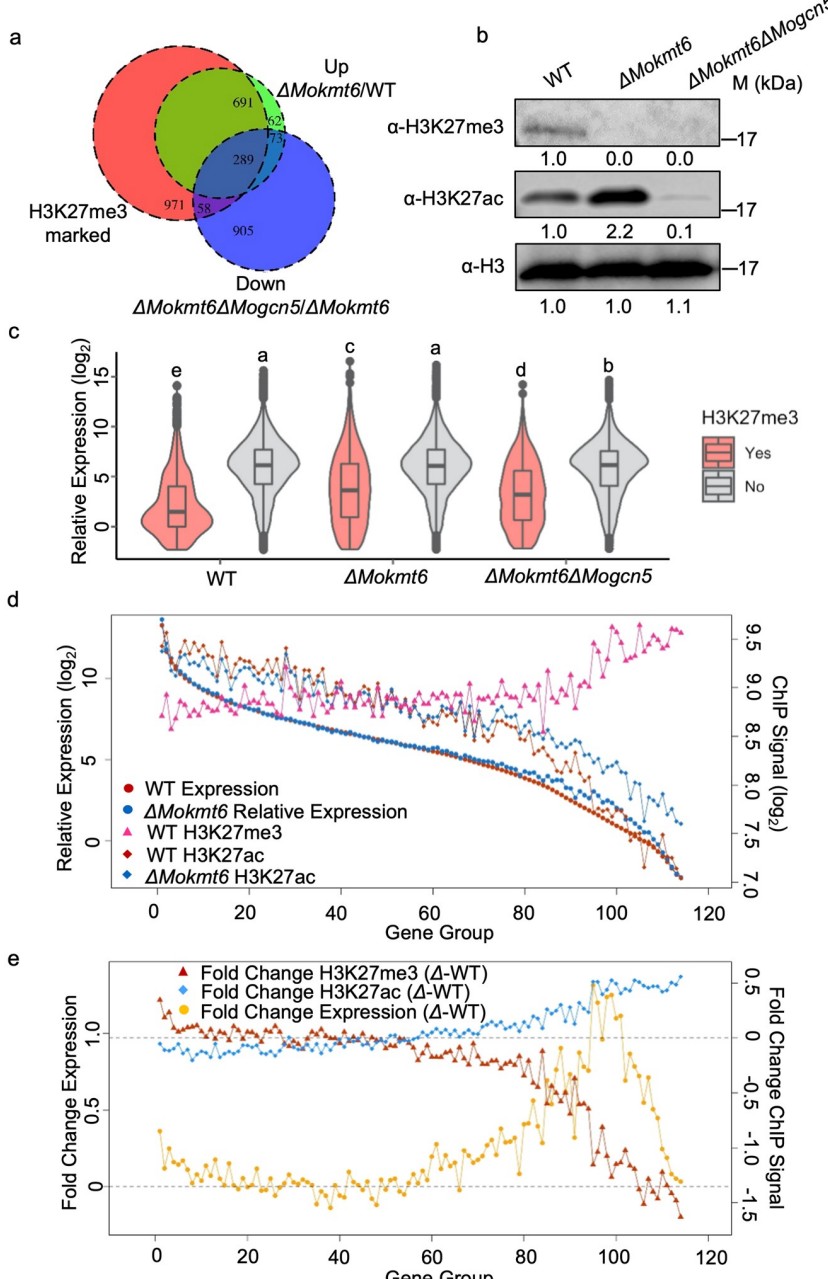

**Fig 4. Loss of trimethylation and gain of acetylation at H3K27 contribute to H3K27me3-marked gene expression.**
(A) Venn diagram illustrating a subset of genes marked by H3K27me3 are up regulated in the mutant *ΔMokmt6*
compared with Guy11 wild type (WT) and some of them are downregulated by further loss of H3K27ac in the double
mutant *ΔMokmt6ΔMogcn5*. All genotypes are grown in liquid complete medium (CM). (B) Western blotting analysis
of protein extracted from wild type (WT), the deletion mutant *ΔMokmt6* with absence of H3K27me3, and the double
deletion mutant *ΔMokmt6ΔMogcn5* with absence of H3K27me3 and reduced of H3K27ac. Blots were probed with the
indicated antibodies. (C) Violin plots showing H3K27me3-marked genes are significantly upregulated in mutants
*ΔMokmt6* and *ΔMokmt6ΔMogcn5*. (D) Genome-wide visualization of relationships between ChIP signals for
H3K27me3/H3K27ac and gene expression across wild type and mutant *ΔMokmt6* abolishing H3K27me3. Total of
11,327 expressed genes out of 12, 101 gene models were divided into 114 groups (100 genes in one group) according to
their expression levels from high to low (x axis). The dots, triangles, and diamonds represent the average levels of gene
expression, ChIP signal for H3K27me3, and H3K27ac in each group, respectively. (E) Genome-wide visualization of
relationships between fold change of ChIP signals for H3K27me3/H3K27ac and fold change of gene expression. The
fold changes of ChIP signals or gene expression were calculated by subtracting the average levels of ChIP signals or
gene expression of individual gene group in wild type (WT) from the mutant *ΔMokmt6* (*Δ*).

there is a clear positive association between increased H3K27me3 and decreased transcription (Fig 4D). This result is consistent with the observation that H3K27me3 is not present uniformly across the genome, but that where present it is associated with repressed transcription. This also shows that the genes with the greatest transcriptional increase in *ΔMokmt6* are those after group 80, where there is clear divergence between the average group RNA-seq levels (Fig 4D, blue versus red circles). These same gene groups are also where the largest bulk redistribution of H3K27ac occur in *ΔMokmt6* compared to wild type (Fig 4D, blue versus red diamonds). Indeed, plotting differential gene expression and fold-change for ChIP signal between wild type and *ΔMokmt6* showed that gene groups having the largest transcriptional activation in *ΔMokmt6* correspond to a concomitant decrease in H3K27me3 and increase in H3K27ac in *ΔMokmt6* (Fig 4E). Collectively, these analyses integrate the transcriptional changes associated with dynamic modifications at H3K27 to detail this control mechanism in *M. oryzae*. The data show that histone modification dynamics at H3K27 provide relatively local (*cis*) transcriptional control.

## H3K27me3 suppresses *in planta* induced genes during *in vitro* growth

Genes required for host infection by *M. oryzae* are generally not expressed during *in vitro* growth [48–50]. As such, another prediction of our hypothesis is that H3K27me3 is present at host induced genes when the fungus is grown *in vitro*, functioning to repress transcription. To assess this, we compared *in vitro* RNA-seq and ChIP-seq results with RNA-seq collected from *M. oryzae* during rice infection. Interestingly, the expression profiles for H3K27me3-marked genes showed a statistically significantly higher *in planta* expression compared to *in vitro* growth, while non-marked genes have a similar expression profile between the two conditions ($p_{marked}$ = 5.7e-14, $p_{non}$ = 0.15, Wilcoxon test, Fig 5A). Among the 1,154 *in planta* up regulated genes, 44.4% were marked by H3K27me3, which is significantly higher than expected given that H3K27me3 marks 16% of genes genome-wide ($\chi^2$ = 939.09, $p < 2.2e-16$, Fig 5B and 5C). Of the *in planta* up regulated genes marked by H3K27me3, two thirds (339/512) were also up regulated in *ΔMokmt6* (Fig 5B). The *in vitro* ChIP-seq profiles were separated based on *in planta* expression profiles. A clear pattern is evident where *in planta* induced genes have the highest average H3K27me3 and H3K36me3 signal, but the lowest for H3K27ac during *in vitro* growth (Fig 5C). Interestingly, the *in planta* downregulated genes showed the highest average H3K27ac signal, but lowest H3K27me3 signal in wild type when grown *in vitro*. The ChIP-seq from *ΔMokmt6* shows that H3K27ac is specifically re-distributed to sites that were previously H3K27me3 and devoid of H3K27ac, while the same loci lose H3K36me3 (Fig 5C). These results collectively show that *in planta* induced genes are significantly enriched for H3K27me3 when grown *in vitro* under transcriptionally repressive conditions. The significant association between in *ΔMokmt6* upregulated genes and those induced *in planta* indicates that the H3K27me3 mark is not just present, but required for transcriptional repression.

## Effectors regulated by H3K27 modification dynamics

To date, numerous *M. oryzae* effectors or presumed effectors have been identified, but little is known regarding their transcriptional regulation [51]. To further test the hypothesis that H3K27 modification dynamics contributes to *in planta* gene induction, we analyzed specific sets of putative and characterized genes contributing to *M. oryzae* host colonization. The effector P software predicted 409 putative *M. oryzae* effectors from our annotated proteins with a secretion signal [52]. Of these, only 349 gene models had read support from our RNA-seq data. We found that 92 of these putative effectors (i.e. 26.4% of the total predicted) are marked by an H3K27me3 domain, which is a higher association than expected by chance ($\chi^2$ = 10.178,

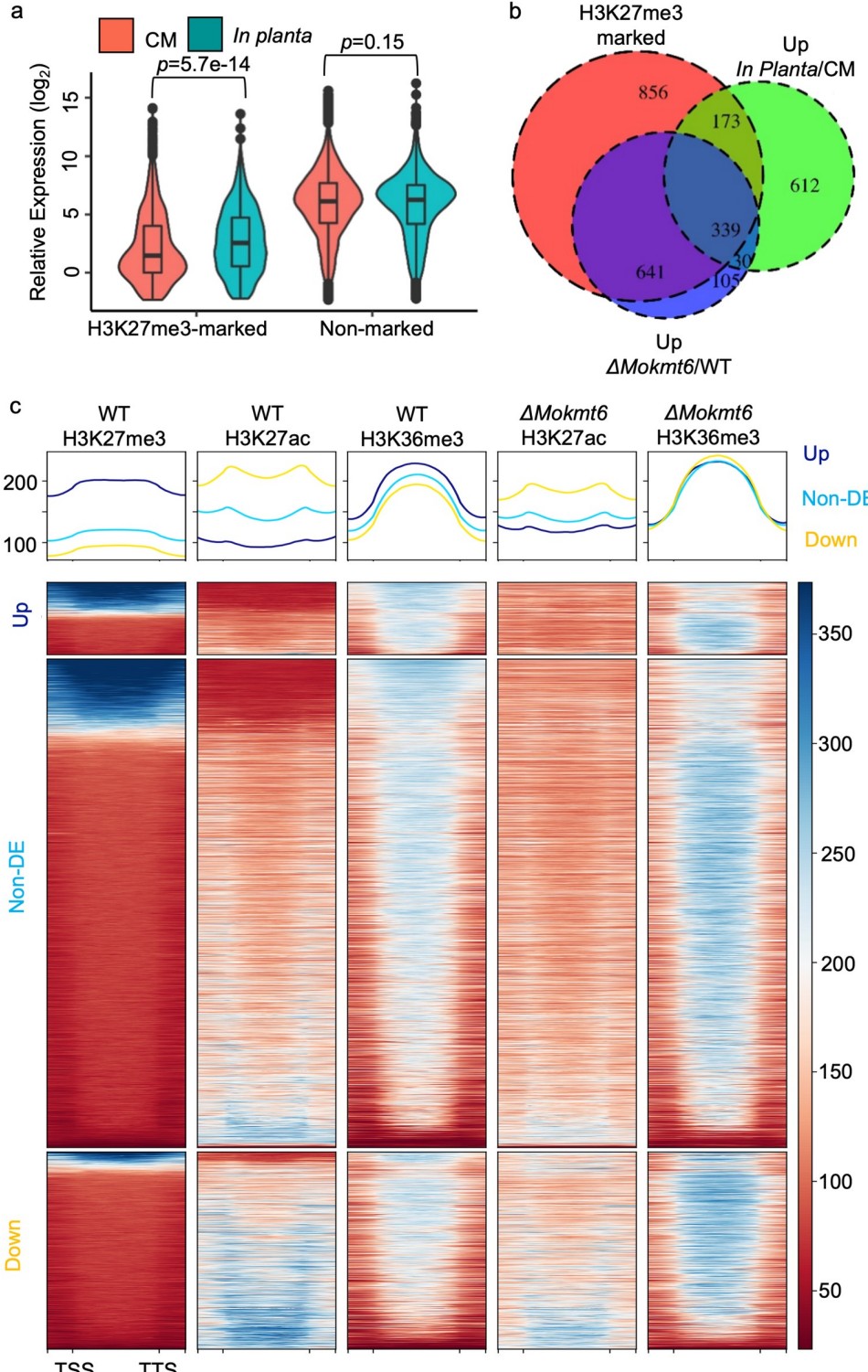

**Fig 5. H3K27me3 marked genes are enriched for *in planta* induced genes.** (A) Violin plots showed H3K27me3 marked genes identified in Guy11 wild type (WT) grown in complete medium (CM) are significantly upregulated *in planta*. (B) Venn diagram illustrating the enrichment of H3K27me3 marked genes in upregulated genes *in planta* and in mutant *ΔMokmt6* abolishing H3K27me3. (C) Heatmaps visualizing ChIP signals for H3K27me3, H3K27ac and H3K36me3 across differential expressed gene sets identified by comparison of relative gene expression from wild type growing *in planta* and liquid complete medium.

$p$ = 0.001422, S7 Table). Further restricting the set of 349 putative effectors to those that are more highly expressed *in planta* versus media, identified 140 genes, of which 50 (35.7%) fall inside an H3K27me3 domain. This represents a highly significant association between *in planta* induced putative effectors and H3K27me3 domains ($\chi^2$ = 45.217, $p < 0.00001$, S8 Table).

As another approach, we curated a list of 31 published effectors and avirulence genes that have direct experimental evidence. Of these well studied effectors and avirulence genes, 27 are present in *M. oryzae* strain Guy11 (S9 Table). We found that over half (14 of 27) of these high-confidence effectors were marked by H3K27me3 during *in vitro* growth, which is significantly higher than expected ($\chi^2$ = 24.285, $p$ = 0.0005, S10 Table). The H3K27me3 marked effector genes include five cloned *Avirulence* (*AVR)* genes, four Biotrophy-associated secreted (BAS) proteins encoding genes, and five genes encoding proteins that can induce or repress pro-grammed cell death (S9 Table) [34,36,48,50,53–60]. Hierarchical clustering based on transcriptional profiles from four conditions identified four group's with similar expression profiles (Fig 6A). Genes in group 1 and 2 are *in planta* induced and transcriptionally responsive to genetic perturbation of H3K27. Specifically, the loss of H3K27me3 significantly increased expression for group 1 and 2 genes, which were dependent on H3K27ac in group 1 and to a lesser extent for group 2 (Fig 6B, 6C, 6F and 6G). Genes in group 3 were also *in planta* induced, but most of them did not show transcriptional regulation by H3K27 modification except for *BAS4* and *MoCDIP9*, while those in group 4 were not shown to be *in planta* induced in our dataset (Fig 6D, 6E, 6H and 6I). Locus specific examples for each group show the dynamics of histone modifications and transcriptional output (Fig 6F–6I and S15 Fig). For example, *SLP1* is present in an H3K27me3 domain, lacking H3K27ac and is not expressed during wild type *in vitro* growth (Fig 6F and S15A Fig). When the *ΔMokmt6* strain is grown *in vitro*, peaks of H3K27ac can be seen at the presumed promoter and potentially 3' regulatory region of the *SLP1* coding sequence along with induced transcription, which is partially lost in *ΔMokmt6Δ-Mogcn5*. The *MoCDIP5* locus showed a similar gene expression and histone modification pattern, by which loss of H3K27me3 results in increased transcription and a concomitant increase in H3K27ac (Fig 6G and S15B Fig). Although H3K27me3 is present at *BAS1*, the loss of the mark did not impact expression in *ΔMokmt6*, suggesting induction of *BAS1* is likely dependent on other unknown transcriptional activators mediated by plant specific or induced signals (Fig 6H and S15C Fig). Of the 27 high-confidence effectors, 12 (44.4%) were induced in *ΔMokmt6* compared to wild type, which is far greater than the genome-wide 9.8% of genes upregulated by *ΔMokmt6*, supporting an association between H3K27me3 and effector regulation ($\chi^2$ = 36.327, $p < 0.00001$, S11 Table). These results support our hypothesis that H3K27me3 is responsible for the repression of many presumed effectors during growth *ex planta*, while H3K27ac can play an important role in activating these genes in the absence of H3K27me3.

To date, there are no published data sets of genome-wide *in planta* ChIP-seq for a filamentous pathogen. This is due to technical limitations of having heterogenous samples where the pathogen (i.e. target genome of interest) is less abundant in the mixed sample by orders of magnitude. Despite this, we conducted *in planta* ChIP using a leaf-sheath assay in which *M. oryzae* spores infect rice leaf sheaths [31,36]. We conducted H3K27me3, H3K27ac, and IgG ChIP from three independent infection samples, where an input fraction was collected after grinding, and then each sample was split into separate fractions for ChIP with one of the three antibodies. Subsequently, qPCR was performed (i.e. ChIP-qPCR) using two primer pairs that were either in the gene body or within the putative promoter (i.e. 200–300 bp upstream of the transcription start site) for two characterized *in planta* induced genes, one control gene

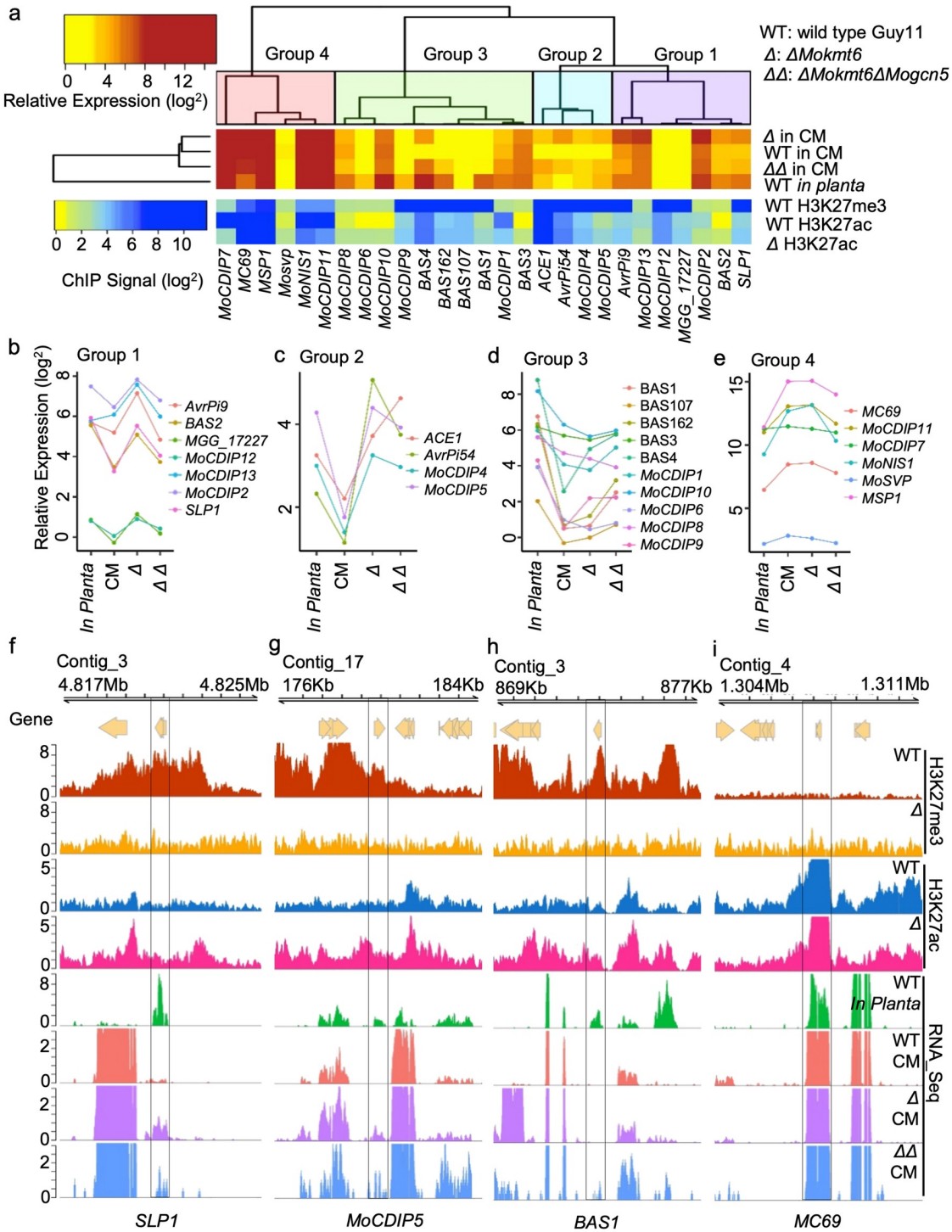

**Fig 6. Fungal effector genes are regulated in H3K27me3-dependent and -independent manners.** (A) The heatmap (top) identifying four groups of 27 characterized effector genes based on their gene expression pattern across three genotypes growing either *in vitro* complete medium (CM) or *in planta*. The heatmap (bottom) showing the status of H3K27me3 and H3K27ac for all effector genes under the same experimental condition as their gene expression. The effector gene expression data were collected by RNA-seq from Guy11 wild type (WT) growing either *in vitro* complete medium or *in planta*, and single deletion mutant *ΔMokmt6* (*Δ*) and double deletion mutant *ΔMokmt6ΔMogcn5* (*ΔΔ*) growing in complete medium. The *ΔMokmt6* lacks H3K27me3 and *ΔMokmt6ΔMogcn5* lacks H3K27me3 and majority of H3K27ac. (B) to (E) showing the expression pattern of effector genes in each group. (F) to (I) showing the profiles of ChIP-seq reads (bins per million mapped reads, BPM) for H3K27me3 and H3K27ac, and RNA-seq reads (BPM) at specific effector gene loci from each of the four groups.

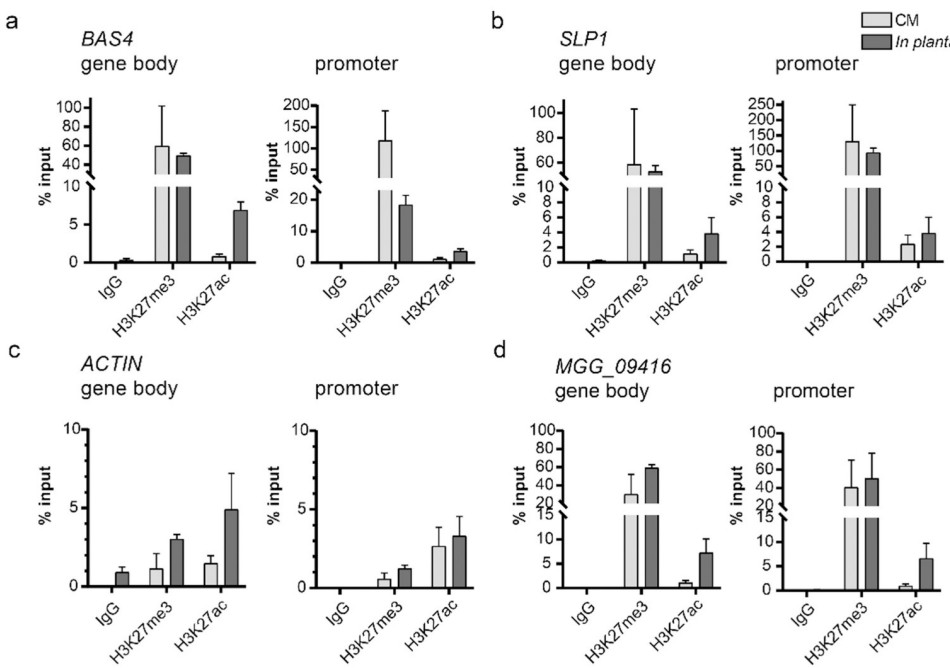

**Fig 7. H3K27 modifications undergo dynamics at specific genomic locations *in planta*.** (A-D) The amount of DNA associated with H3K27me3 and H3K27ac was determined relative to the amount of input DNA (% Input) from *M. oryzae* during axenic growth (CM) and during plant infection (*in planta*). A control ChIP sample was conducted where the IgG antibody was used for the pull-down (IgG sample) and did not produce a significant signal for any primer pairs tested. For each gene locus, one primer pair was designed in the gene body while the was designed in the promoter (i.e. 200–300 bp in front of the transcription start site). (A) *BAS4* locus, (B) *SLP1* locus, (C) *ACTIN* locus, and (D) *MGG_09416* locus.

(*Actin*) that lacked H3K27me3 during axenic growth, and another control (*MGG_09416*) which contained H3K27me3 during axenic growth. Results from the *in planta* ChIP-qPCR were compared to ChIP samples from axenic growth. Our results demonstrate that H3K27me3 was highly abundant and at comparable levels between *in plant* and *axenic* growth for the gene body location tested for *BAS4*, *SLP1* and control gene (*MGG_09416*) (Fig 7A, 7B and 7D). Interestingly, the H3K27me3 levels dropped 6-fold at the *BAS4* promoter during *in planta* infection compared to axenic growth, while the levels remained unchanged for *SLP1* and *MGG_09416* (Fig 7A, 7B and 7D). We did observe small changes for H3K27me3 levels between growth conditions at the gene body and promoter for the *Actin* locus, but the amount of H3K27me3 was low, consistent with ChIP-seq data (Fig 7C). We did observe a large increase in H3K27ac levels during *in planta* growth at the gene body of *BAS4*, but we also observed this at the *Actin* and *MGG_09416* loci, which are not differentially expressed between the growth conditions. These results are consistent with our genome-wide analyses, in which H3K27ac was associated with increased expression, but it was not absolutely sufficient or required (e.g. S9 Fig). The control ChIP-qPCR, where IgG was used for the pull-down, did not produce significant qPCR signal, indicating the histone-modification antibody pull-down was specific (Fig 7). Overall, these results demonstrate dynamics for H3K27 modification at specific well characterized genes using both genetic perturbation and *in planta* infection experiments. These data provide important mechanistic information regarding the repression and activation of *in planta* induced genes.

## Discussion

The aim of this research was to further understand the transcriptional regulation of *in planta* induced genes in *M. oryzae* by investigating the role of histone modification dynamics at H3K27. Our integrated multi-omics and molecular genetic investigation provides strong evidence that H3K27me3 is an important and specific negative regulator of *in planta* induced genes when *M. oryzae* is grown *ex planta*. We found that only 16.6% of genes were marked by H3K27me3, yet 44.4% of *in planta* induced genes were H3K27me3 marked, a striking enrichment. Not only is the mark present at many *in planta* induced loci, it appears to play a functional role based on the significant overlap between *in planta* induced genes and those upregulated in *ΔMokmt6*, including many known or presumed effectors (i.e. *BAS4*, *BAS2*, *AVR-Pi9*, *SLP1*). This is consistent with reports that H3K27me3 represses transcription of specific secondary metabolite clusters in *Fusarium* species, and also the endophyte *E. festucae*, where knockout or knockdown of H3K27me3 results in transcriptional activation [10,19–21]. Secondary metabolite clusters have similar biological function to effectors in that they are transcriptionally repressed outside of specific growth conditions, they function outside of the producing organism, and they serve adaptive functions [61,62]. As such, there is a clear role of H3K27me3 playing a conserved role in silencing adaptive genes (i.e. effectors and secondary metabolite clusters) in pathogenic fungi, but further mechanistic understanding of the dynamics, signals and enzymes in this pathway are required.

The clear boundaries between H3K27me3 and H3K27ac domains in *M. oryzae*, and their association with TEs versus genes respectively, highlight their distinct roles in transcriptional regulation. In *F. fujikuroi*, a global increase in H3K27ac was reported when H3K27me3 was reduced by RNAi, but the alterations were not analyzed at specific loci [19]. Here we present detailed analysis showing the specific re-distribution of H3K27ac in the absence of H3K27me3 is at local sites that were previously trimethylated, and not a general genome-wide increase of H3K27ac. Importantly, a significant amount of the upregulated genes in *ΔMokmt6* showed lower transcription in *ΔMokmt6ΔMogcn5*. There is a confounding effect of GCN5 because it can acetylate additional lysine residues besides H3K27 [45]. Our results demonstrate a GCN5 dependence for some transcriptional activation in the absence of H3K27me3 (~1/3), but this could be caused by additional means beyond H3K27ac. The specific accumulation of H3K27ac at previously H3K27me3 marked regions in *ΔMokmt6* suggests a regulatory role, but further evidence is required. What protein(s) or signal recruits GCN5 to sites of previous H3K27me3 marked regions is not clear. Nor is it known how H3K27me3 is established at repressed sites, prior to and excluding H3K27ac. A number of *in planta* induced genes were not responsive to genetic perturbation of H3K27 modification, and further research is needed to test whether H3K27 modification dynamics cooperating with transcription factors contribute to *in planta* induced gene regulation in *M. oryzae*.

Similar to H3K27ac, H3K36me3 modification is thought to antagonize H3K27me3 in many systems and is associated with active transcription [43,44]. Our result show H3K36me3 is nearly ubiquitous for genes in *M. oryzae*, regardless of transcription, which was also found in *F. graminearum* [10]. However, in *F. graminearum* there was no interaction observed for H3K36me3 when H3K27me3 was genetically perturbed. In *M. oryzae*, we found H3K36me3 levels were significant reduced in *ΔMokmt6* at loci that had H3K27me3 in the wild type, indicating a positive association between the presence of the marks. In *N. crassa*, the histone methyltransferases NcSet2 catalyzes the majority of H3K36me3, while, another methyltransferases NcAsh1 deposits only 5% of H3K36me3 [63]. Interestingly, NcAsh1 deposited H3K36me3 is associated with lowly transcribed genes and domains of H3K27me2/3. Unlike our findings, the level of NcAsh1 catalyzed H3K36me3 did not change in the absence of

H3K27me2/3 [63]. Additionally, in *F. fujikuroi*, ASH1 was responsible for a minor portion of genome-wide H3K36me3, but ASH1-mediated H3K36me3 was associated with regions enriched for H3K27me3 [64]. Interestingly, ASH1 has a reported role in genome stability, possibly through DNA repair, in *F. fujikuroi* [64]. The role of SET2 and ASH1 in catalyzing H3K36me3 in *M. oryzae* is not clear [39,65], but our results are consistent with *M. oryzae* also having two distinct chromatin states associated with H3K36me3. As suggested, it is an interesting hypothesis that distinct complexes may deposit H3K36me3 and H3K27me3 at specific facultative heterochromatin sites for a specific regulatory mechanism [63,64]. It will be interesting to determine if altered H3K36me3 distribution in the absence of H3K27me3 is SET2- or ASH1-dependent.

Our research specifically addressed the hypotheses that H3K27me3 represses transcription during *ex planta* growth, and that switching from a trimethylated to acetylated state at H3K27 underlies transcriptional activation at specific loci. While it is still not technically feasible to address histone modification dynamics at a global scale during plant infection, we sought to address predictions of this hypothesis using genetics and *in vitro* growth. For this model, we show that PRC2-mediated H3K27me3 represses transcription of many *in planta* induced genes, H3K27 modification dynamics from trimethylation to acetylation contribute to this transcriptional activation, and finally we report an association between H3K27 modification-mediated transcriptional changes and genes highly transcribed *in planta*. In addition, we report high-quality data from successful *in planta* ChIP-qPCR. These results showed that H3K27me3 levels were generally stable between *in planta* and axenic growth at many tested genomic sites. We also detected a significant reduction at an effector promoter, where the level of H3K27me3 was reduced 6-fold compared to axenic growth. Our results show that histone modification status can be significantly altered *in planta*, but our analysis is limited to a few loci. Questions remain regarding how H3K27 modification dynamics, their crosstalk with additional histone modifications (e.g. H3K36me3) and their impact on DNA accessibility change across growth conditions. Technological advances are required to conduct genome-wide ChIP and chromatin analysis during *in planta* infection to more completely understand transcriptional regulation during host infection. Our observations support the hypothesis that modification dynamics at H3K27 is an important regulatory mechanism for host-adapted genes in *M. oryzae*.

## Methods

### Fungal strains and incubation condition

*M. oryzae* wild type strain Guy11 (French Guiana) was used throughout this study. Complete medium agar/broth was used for testing vegetative growth and preparing materials for DNA, RNA and protein extraction [66]. Rice polish agar (rice polish 25 g/L, 16 g/L agar, autoclaved for 46 mins) was used for testing vegetative growth and conidiation. TB3 medium (Sucrose 200 g/L, Casamino acid 6 g/L, yeast extract 6 g/L, agar 15 g/L) was used for screening fungal transformants. For test vegetative growth, *M. oryzae* strains were grown on complete medium agar plates at 28˚C in the dark, and colony diameters were measured after 12 days of growth. *M. oryzae* strains were grown on rice polish agar plates at 28˚C for the first 7 days then cultured under light for the remaining 5 days. For conidiation, conidia were collected from rice polish agar plates under constant light for 10 days by using 5 mL sterilized water. The conidia morphology was checked with Zeiss Axioplan 2 IE MOT microscope. For preparing materials for DNA, RNA and protein extraction, 5 to 7 pieces of fungal plugs from rice polish agar were cultured in the liquid complete medium for 3 to 4 days at 28˚C under dark conditions.

## Plasmid construction and fungal transformation

The single deletion mutants for *MoKmt6*, *MoSuz12*, *MoEed*, *MoGcn5* and double deletion mutants for *MoKmt6/MoGcn5* were acquired by a homologous recombination strategy. Briefly, approximately 1.0 kb upstream and 1.0 kb downstream sequences of each gene were amplified from wild type Guy11 genomic DNA and fused with the hygromycin resistance cassette or G418 resistance cassette in binary vectors pFGL821 or pFGL921. pFGL821 was a gift from Dr. Naweed Naqvi (Addgene plasmid # 58223), and pFGL921 was modified from pFGL821 by replacing the hygromycin resistance cassette with a G418 resistance cassette. Drug resistant transformants were screened by PCR. The primers used in this study are listed in S6 Table.

For gene complementation assays, entire *MoKmt6*, *MoSuz12* and *MoEed* genes with upstream about 1.5 kb potential promoter and downstream about 0.5 kb potential terminator were amplified and cloned into binary vector pFGL921 with the G418 resistance cassette. After Sanger sequencing, confirmed corresponding plasmids were transferred into each individual deletion mutant. G418 resistant transformants were screened by PCR.

*Agrobacterium tumefaciens*-mediated transformation was performed as previously described with minor modifications [67]. Briefly, *M. oryzae* strains were cultured in growth chamber at 25°C under light for 2 to 3 weeks. Spores were collected at the final concentration of $2x10^6$ spores/mL. A single colony of *Agrobacterium* strain AGL1 with appropriate binary vector was mixed with around 100 μL of above spore solution and co-cultivated on a mixed cellulose ester membrane (GE Healthcare, # 09-529-711) for 48 hours. After co-cultivation, the membrane was transferred to TB3 with 400 mM cefotaxime and appropriate fungal selections (200 μg/mL hygromycin (Corning, # MT30240CR) or 800 μg/mL G418 (VWR, #97064–358)). After 8 to 12 days' culture, the fungal antibiotic resistant colonies were transferred to individual complete medium plates with appropriate antibiotic (200 μg/mL hygromycin or 500 μg/mL G418) for further screening. The fungal colonies with hygromycin or G418 resistance were screened by PCR analysis to validate the successful transformants.

## Plant materials, leaf infection and leaf sheath assay

Susceptible rice cultivar YT16 was used in this study for infection assay. Six to seven seeds were sown in one pot filled with Baccto soil. Three-week old rice was used for inoculation. For rice leaf infection assay, $5x10^4$ spores/mL was collected from related strains through two-layers of miracloth with 0.25% gelatin. The spore spray inoculation, further incubation and disease severity were performed as previously described [68]. Two biological replications were performed for each strain and control (0.25% gelatin). The rice leaf sheath assay was conducted as previously described to collect samples for *in planta* RNA sequencing analysis [36]. In brief, the rice leaf sheath was cut into 7 cm long sections with forceps and inoculated with around 80 μL of fungal spore solutions at the concentration of $1x10^5$ spores/mL. Wild type Guy11 spores were harvested from about 3-week old plates with 0.25% gelatin solution for preparing spore suspensions. The inoculated rice leaf sheaths were supported horizontally in tray containing 1.5% agar such that the spores settled on the mid vein regions. The tray was placed at 28°C in a growth chamber in a dark. At 36 hours post inoculation, the sheaths were hand-trimmed to remove the sides around the mid vein to enrich the fungal biomass. Three replicates of infected rice leaf sheath were collected at 36 hours post inoculation for RNA sequencing analysis or ChIP-qPCR assay.

## Western blotting assay

All strains were cultured in liquid complete medium with 150 rpm shaking at 28°C for 2–3 days. About 200 mg of fungal mycelia were harvested and homogenized with beads mill

homogenizer (OMNI international). The homogenized tissues were resuspended in 600 μL protein lysis buffer (50 mM Tris-HCl, pH 7.5, 100 mM NaCl, 5 mM EDTA, 1% Triton X-100, 2 mM Phenylmethanesulfonyl fluoride, 100 μM Leupeptin, 1 μg/mL Pepstatin, 10 μM E-64). The lysates were centrifuged at 4˚C at 12,000 rpm for 20 min. The supernatant containing proteins were collected in a fresh tube, mixed with 4x protein loading buffer (200 mM Tris-HCl (pH 6.8) 400 mM DTT, 8% SDS, 0.4% bromophenol blue, 40% glycerol) and denatured at 95˚C for 5 min. The proteins were separated on 4–20% precast protein gel (Bio-Rad) and transferred onto PVDF membranes. Western blotting was performed by by iBind western system (Thermo Fisher) followed manufacturer's protocol. Primary antibodies including anti-H3K27me3 (Diagenode, #C15200181,1:1000 or 1:500) anti-H3K27ac (Abcam, #ab4729, 1:20000), and anti-H3 (Sigma, #06–755, 1:500) with secondary antibodies including Goat Anti-Mouse IgG H&L (HRP) (Abcam,#ab6789, 1:2000) or Goat Anti-Rabbit (HRP) (Epigentek,#A12004-1, 1:1000) were used for immunoblotting. Amersham ECL Prime Western Blotting Detection Reagent (GE Healthcare, #45-002-401) was used for signal detection. All Western blot assays were performed with at least two biological replicates and quantified by ImageJ (https://imagej.nih.gov/ij/).

## RNA extraction, cDNA synthesis and RT-PCR

RNA extraction was performed using TRIzol (Invitrogen) following manufacturer's protocol. RNA quality was determined by NanoDrop spectrophotometer (Thermo Fisher). 1 μg of total RNA was used for cDNA synthesis by M-MuLv reverse transcriptase (Thermo Fisher) following the manufacturer's protocol.

## ChIP sequencing

For ChIP-seq, samples from *M. oryzae* wild type and mutant *ΔMokmt6* grown in liquid complete medium were collected. Three independent biological replicates were performed for each sample in the same experiment. The ChIP procedure was performed as previously described with a modification [69]. In brief, ChIP was performed 3-day-old fungal mycelia growing in liquid culture of CM. Fungal mycelia were washed twice in phosphate-buffered saline (PBS, Sigma) and dried by sterilized paper. For cross-linking, dried fungal mycelia were then transferred into a 50 mL fresh tube and incubated with 30 mL of 1% formaldehyde at room temperature on a shaker for 30 min. The reaction was quenched by adding 10% of 2.5 M glycine and incubating for 5 min. The cross-linked mycelia were washed with cold PBS for three times, dried, and flash frozen in liquid culture and stored at -80˚C for further ChIP assays.

Total of 0.1 g of cross-linked fungal mycelia was used for each ChIP assay reaction. The desired amounts of fixed mycelia were transferred into 1.5 mL sonication-specific tube (Diagenode) and resuspended in lysis butter with phosphatase inhibitor cocktail (Sigma). Cell lysis and chromatin shearing was conducted by two rounds of sonication (10 cycles, 30"on/30" off) on a Biorupter (Diagenode) at 4˚C. Lysates were centrifuged at 13,000 rpm for 10 min at 4˚C. The MNase treatment was followed by adding final concentration of 2 unit of MNase and incubating at 37˚C for 15 min. The stop buffer of 0.5 M EDTA and 4M NaCl was added to quench the MNase digestion. The supernatant was collected by centrifuge and 20 μL of cell lysate was saved as input sample. A pre-clear procedure was conducted by incubating each sample with 20 μL of SurebBeads protein A/G magnetic beads (1:4, v/v) (Bio-rad) for one hour to remove non-specific binding proteins. For detection of H3K27me3, H3K27ac, and H3K36me3, 2% (v/v) of antibodies of anti-H3K27me3 (Abcam, #ab6002), anti-H3K27ac (Epigentek, #A-4708-100), and anti-H3K36me3 (Abcam, #ab9050) was added to individual reactions, respectively. Following overnight incubation, protein A/G magnetic beads of 40 μL were

added and incubated for three hours to bind the antibody. The beads with bound chromatin were then pelleted by magnetic rack, and washed twice by lysis buffer, once by high salt buffer, and finally by TE buffer.

Chromatin de-crosslinking and DNA purification were conducted for both input and anti-body pulled down DNA by iPure Kit v2 (Diagenode). To obtain the insert DNA at a size of 150 to 450 bp, the purified DNA was sized selected by AmPure beads (Beckman Coulter) according to the dual size selection protocol. At least 4 ng of DNA from each reaction was used as input for library preparation. In brief, the DNA was first end repaired and A-tailed using NEB-Next End Repair Module kit (NEB), then followed by universal dual adapter ligation (IDT). Adapter-ligated DNA fragments were enriched by PCR using Phusion Master Mix (NEB) and primers unique containing barcoding information (IDT). The PCR program was set for 8 cycles to reduce the bias caused during amplification. The cleaning up procedure was performed after each enzyme treatment by 0.8x AmPure beads in high-salt purification buffer or purification buffer. Successfully amplified libraries were size-selected and cleaned up by 0.8X AmPure beads to get rid of primer-dimer contaminants. There are total of 24 ChIP-seq libraries with three independent biological replicates for each sample in this study. All libraries were pooled for Illumina sequencing. Single-end 75-bp sequencing was conducted using an Illumina NextSeq500 Instrument at the Integrated Genomics Facility of Kansas State University.

### *In planta* ChIP-qPCR

The infected rice leaf sheath samples for ChIP-qPCR assay were first performed the ChIP procedure as described in ChIP sequencing with a slight modification. After cross-linking, we combined five pieces of infected rice leaf sheathes as one biological replicate and fine ground the tissue in liquid nitrogen before adding the ChIP buffer. The purified input and antibody pulled down DNA were used for ChIP-qPCR assay. ChIP-qPCR was performed in CFX 96 Real-Time PCR system (BioRad), three biological replicates with two technical replicates for each primer pair were performed with SYBR Select Master Mix for CFX (Applied Biosystems). The length of all designed primer pairs was ranged from 80–120 bp, as listed in S6 Table ChIP-qPCR data was normalized with 'Percent Input Method' as described in (https://www.thermofisher.com/us/en/home/life-science/epigenetics-noncoding-rna-research/chromatin-remodeling/chromatin-immunoprecipitation-chip/chip-analysis.html).

### Stranded RNA sequencing

The infected rice leaf sheath samples for RNA sequencing analysis were flash frozen in liquid nitrogen and stored at -80˚C for further total RNA extraction and library preparation. Stranded RNA-seq library preparation was mainly based on previous methods and TruSeq stranded mRNA sample prep guide from Illumina [70–72]. After total RNA extraction, mRNA was isolated from total RNA using the Dynabeads Oligo(dT)$_{25}$ mRNA Purification Kit (Thermo Fisher). First-strand DNA/RNA hybrid was created by the SuperScript III kit (Thermo Fisher) with random primers and RNAseOut (Thermo Fisher). Second-strand cDNA was synthesized by DNA Polymerase I (Thermo Fisher) and RNase H (NEB) in the second-strand buffer using dNTPs with dTTP replaced by dUTP. The double-strand cDNA was then fragmented using the Fragmentase enzyme (NEB) and used as input for library preparation as described above except 14 PCR cycles for amplification. There were total of twelve RNA sequencing libraries accounting for four samples with three independent biological replicates, including one Guy11 wild type strain, one single mutant *ΔMokmt6#77* strain, and one double mutant *ΔMokmt6ΔMogcn5#18* strain under *in vitro* liquid complete medium culture condition and one Guy11 wild type strain under *in planta* condition. Single-end 75-bp

sequencing was conducted using an Illumina NextSeq500 Instrument at the Integrated Genomics Facility of Kansas State University.

## ChIP-seq and RNA-seq data analysis

The quality of individual Fastq files derived from Illumina sequencing was checked by FASTQC (version 0.11.3; www.bioinformatics.babraham.ac.uk/projects/). Short reads were trimmed by TrimGalore (https://github.com/FelixKrueger/TrimGalore) to drop off the low-quality reads. Short reads for ChIP-seq and RNA-seq were mapped using Burrows-Wheeler Aligner (BWA) or STAR for ChIP-seq or RNA-seq, respectively, to the latest *M. oryzae* Guy11 genome annotated by our lab [73–75]. The BWA-mem algorithm used for ChIP-seq alignments with default parameters. For STAR RNA-seq mapping, we used the output from—*quantMode GeneCounts* to summarize counts over gene features which only reports uniquely mapped reads. Peak calling was conducted using "callpeak" function in MACS2 [76]. ChIP-seq reads coverage was averaged, normalized and analyzed using deepTools [77]. ChIP-seq summary was conducted by bedtools. The called peaks were used to identify association to features, such as genes and TEs, by identifying where the peaks and the features overlap. Genes and TEs that were overlapped by a ChIP-peak for at least 50% of the gene or TE length were counted as 'marked' by that histone modification. Pearson correlation analysis was conducted to test the ChIP-seq data reproducibility among three biological replicates (S16 Fig). Principal component analysis was conducted to investigate gene expression profiles at the whole genome level among different genotypes growing under different conditions with three biological replicates (S17 Fig). Gene differential expression was identified using DESeq2 [78]. Genes were determined as differentially expressed if their expression had a minimum 2-fold changes with a statistical significance $p < 0.05$. The comparison was conducted on wild type under complete medium and *in planta* growth conditions or between wild type and mutants under complete medium growth conditions. We conducted genome-wide effector gene prediction by EffectorP 2.0 [52]. For functional analysis, we conducted gene ontology analysis by Blast2GO [79] and Kyoto Encyclopedia of Genes and Genomes (KEGG) pathway analysis for interested gene sets [80].

## Phylogenetic and domain structure analysis

The protein sequences of potential PRC2 components in filamentous fungi were retrieved from FungiDB data base (https://fungidb.org/fungidb/). The MEGA X was used for sequence alignment and neighbor-joining tree generation with 1000 bootstrap replications [32]. The domain structure was analyzed by pfam (https://pfam.xfam.org/) and visualized by TBtools (https://github.com/CJ-Chen/TBtools/releases) [81].

## Statistical analysis

All statistical analysis was conducted using the R version 3.5.3 statistical environment (R Core Team, 2019). Chi-square test was conducted using chisq.test() function from MASS package [82]. Multiple comparison was conducted by HSD.test() function from agricolae package [83]. Significant enrichment of overrepresented functional categories was determined using Fisher's exact test ($p < 0.05$) with adjusted $p$ values by Benjamini–Hochberg procedure option. The ggplot2 package was used for data visualization [84]. The Gviz package was used to visualize genome data [85]. Heatmap plots were generated by heatmap3 package [86].

## Supporting information

**S1 Fig. Distribution of H3K27me3, H3K27ac, and H3K36me3 in *M. oryzae* Guy11.** (A) and (B) Violin plots illustrating genome-wide distribution of domains (A) and ChIP signals (B) of H3K27me3, H3K27ac, and H3K36me3 in Guy11 wide type growing under *in vitro* complete medium. Letters above the violin plots indicates the significance.
(TIF)

**S2 Fig. Distribution of histone modifications at H3K27 across diverse types of transposable elements in *M. oryzae*.** (A) and (B) ChIP signals of H3K27me3 and H3K27ac from MACS2 peak calling across transposable elements (TE) families (A) and subfamilies (B). ChIP-Seq data were collected from *M. oryzae* Guy11 wild type growing under *in vitro* complete medium. The number of TEs for each group is shown above each violin plot. Letters above the violin plots indicate the significant difference among groups based on ANOVA and Tukey's HSD test.
(TIF)

**S3 Fig. Phylogenetic and homology matrix analysis of PRC2 core components in different organisms.** (A) Neighbor-joint tree of selected PRC2 core components generated by MEGA X with 1000 bootstrap replications. The protein domains are predicated by SMART and visualized by TBtools. The sequences used for analysis included *MoKmt6* (MGG_00152), *MoSuz12* (MGG_03169) and *MoEed* (MGG_06028) from *M. oryzae*; *FgKmt6* (FGSG_15795), *FgSuz12* (FGSG_04321), and *FgEed* (FGSG_15909) from *F. graminearum*; *FfKmt6* (FFUJ_00719), *FfSuz12* (FFUJ_09784) and *FfEed* (FFUJ_12272) from *F. fujikuroi*; *NcSet7* (NCU07496), *NcSuz12* (NCU05460), and *NcEed* (NCU05300) from *N. crassa*; *Ezh2* (NP_001190176.1), *Ezh1* (NP_001308008.1), *Suz12* (NP_056170.2) and *Eed* (AAC23685) from *H. sapiens*; *E(z)* (NP_001137932.1), *Su(z)12* (NP_730465.1), and *Esc* (NP_477431.1) from *D. melanogaster*. (B) Homology matrix analysis of *M. oryzae* PRC2 core components with PRC2 homologs in different organisms as described above. Numbers indicate protein coverage and identity.
(TIF)

**S4 Fig. Homologous recombination strategy was used for generating deletion mutants.** (A) Homologous recombination was used for gene knockout. ±1 kb gene flanking region (black lines) was amplified for targeting gene of interest. The gene coding region was replaced with hygromycin resistant cassette (HPH) for single knockout or geneticin resistant cassette (G418) for double knockout. Inside primer pair (blue arrow) was used for testing the presence/absence of gene of interest in transformants. Outside primer pair (yellow arrow) was used for testing whether there is correct resistant cassette integration in transformants. Green lines indicate upstream and downstream sequences of interested gene and red lines indicate upstream and downstream sequences of the resistant cassette. (B) *ΔMokmt6*, *ΔMoeed*, and *ΔMosuz12* were confirmed by PCR amplifications with inside and outside primers.
(TIF)

**S5 Fig. The complemented strains were screened by PCR.** The complementation for *ΔMokmt6*, *ΔMosuz12* and *Δmoeed* was screened by inside and outside primers. Amplifications with inside primers suggested the success in re-introducing the original gene back to the deletion mutants.
(TIF)

**S6 Fig. Loss of H3K27me3 leads to dynamics of H3K27ac and H3K36me3 at whole genome level.** Heatmap illustrating pairwise Pearson correlation coefficiency of ChIP signals of H3K27me3, H3K27ac, H3K36me3 between Guy11 wild type (WT) and mutant *ΔMokmt6*

lacking H3K27me3.
(TIF)

**S7 Fig. Loss of H3K27me3 leads to re-distribution of H3K27ac and H3K36me3.** Heat maps visualizing profiles of ChIP signals for H3K27me3, H3K27ac and H3K36me3 across transposable elements (TEs) in Guy11 wild type (WT) and mutant *ΔMokmt6*. Two clusters of TEs are generated by unsupervised k-means analysis and ranked according to their H3K27me3 enrichment from wild type grown *in vitro* complete medium.
(TIF)

**S8 Fig. The majority of downregulated genes in ΔMokmt6 are not marked by H3K27me3.** Venn diagram showing the overlap of genes downregulated in ΔMokmt6 and marked by H3K27me3 in Guy11 wild type.
(TIF)

**S9 Fig. Expression profiles in wild type and Δ*Mokmt6* for genes that gained and lost H3K27ac in Δ*Mokmt6*.** (A) Violin plot and (B) scatter plot showing gene expression profiles during *in vitro* complete medium (CM) growth in wild type and Δ*Mokmt6* for 218 expressed genes that gained H3K27ac in Δ*Mokmt6*. (C) Violin plot and (D) scatter plot showing gene expression profiles during *in vitro* complete medium (CM) growth in wild type and Δ*Mokmt6* for 794 expressed genes that lost H3K27ac in Δ*Mokmt6*. Letters above the violin plots indicate the significant difference among groups based on ANOVA and Tukey's HSD test.
(TIF)

**S10 Fig. Expression profiles in wild type and Δ*Mokmt6* for genes that lost H3K36me3 in Δ*Mokmt6* mutant.** (A) Violin plots and (B) scatter plot showing the expression profiles during *in vitro* complete medium (CM) growth in wild type and Δ*Mokmt6* for 321 expressed genes that lost H3K36me3 in Δ*Mokmt6*. Letters above the violin plots indicate the significant difference among groups based on ANOVA and Tukey's HSD test.
(TIF)

**S11 Fig. Comparison of gene expression profiles under a combination of genotypes and growth conditions.** (A) Venn diagram showing the number and overlap of genes that only gained H3K27ac, only lost H3K36me3 or had both changes occur in the Δ*Mokmt6* mutant compared to wild type under *in vitro* complete medium (CM) growth. (B) Violin plots for the RNA-seq expression of wild type (WT_CM) and Δ*Mokmt6* (Δ*Mokmt6*_CM) during *in vitro* complete medium (CM) growth for the gene sets from (A). Letters above the violin plots indicate the significant difference among groups based on ANOVA and Tukey's HSD test.
(TIF)

**S12 Fig. MoGcn5 is involved in mediating majority of H3K27ac in *M. oryzae*.** (A) Δ*Mogcn5* and Δ*Mokmt6*Δ*Mogcn5* were confirmed by PCR amplification with inside and outside primers. (B) The anti-H3K27ac was used for detecting the global level of H3K27ac in the samples and anti-H3 was used as loading control. Signal intensities were measured by ImageJ.
(TIF)

**S13 Fig. The double mutant Δ*Mokmt6*Δ*Mogcn5* has abnormal growth and conidia morphology.** (A) Colony morphology of wild type and Δ*Mokmt6*Δ*Mogcn5* on complete medium agar (CM) at 12 days. (B) Colony diameters measured at 12 days. The growth rates were determined to be significantly different (**, $p < 0.01$) between wild type and Δ*Mokmt6*Δ*Mogcn5* compared using student's t-test. (C) Representative conidial morphology of wild type (WT), Δ*Mokmt6*, Δ*Mogcn5*, and Δ*Mokmt6*Δ*Mogcn5* collected after growth on rice polish agar.

Bar = 20 μm.
(TIF)

**S14 Fig. Single mutants *ΔMokmt6*, *ΔMogcn5* caused reduced symptoms on rice, while *ΔMokmt6ΔMogcn5* fails to cause disease.** (A) Representative blast lesions of rice leaves sprayed with wild type (WT), *ΔMokmt6*, *ΔMogcn5*, *ΔMokmt6ΔMogcn5* and 0.25% gelatin (control). The photos were taken at 7 days post inoculation and show typical blast lesions during a compatible interaction. (B) Bar plots showing quantitative analysis of 14 independent infected rice leaves (n = 14) from two biological experiments, classified as described in [68]. The disease severity rating for each treatment was compared to the wild type infection and student's t-test was used to determine statistically significant differences. **, $p < 0.01$; *, $p < 0.05$.
(TIF)

**S15 Fig. Distinct gene expression pattern of effectors in H3K27me3-dependent and -independent manners.** Expression of effector gene *SLP1* (A), *MoCDIP5* (B), *BAS1* (C), and *MC69* (D). RNA-seq data are collected from wild type (WT) *M. oryzae* strain Guy11 growing under *in vitro* complete medium (CM) and *in planta*, and two mutants *ΔMokmt6* and *ΔMokmt6ΔMogcn5* growing in CM. The mutant *ΔMokmt6* lacks H3K27me3 and the double mutant *ΔMokmt6ΔMogcn5* lacks H3K27me3 and majority of H3K27ac. Letters above the violin plots indicate the significant difference among groups based on ANOVA and Tukey's HSD test.
(TIF)

**S16 Fig. Reproducibility of ChIP-seq data across biological replicates by pair-wise Pearson correction analysis.** ChIP-seq experiment were conducted on *M. oryzae* Guy11 wild type (WT) and the mutant *ΔMokmt6* with absence of H3K27me3 growing in complete medium (CM). R represents biological replicates.
(TIF)

**S17 Fig. Principal component analysis on RNA-Seq dataset.** RNA-Seq experiment was conducted on RNAs extracted from *M. oryzae* Guy11 wild type (WT) growing under *in vitro* complete medium (CM) and *in planta*, and two mutants *ΔMokmt6* and *ΔMokmt6ΔMogcn5* growing in complete medium. The mutant *ΔMokmt6* lacks H3K27me3 and the double mutant *ΔMokmt6ΔMogcn5* lacks H3K27me3 and majority of H3K27ac.
(TIF)

**S1 Table. Summary of ChIP-seq data from experiments conducted on wild type (WT) *M. oryzae*, strain Guy11 and mutant *ΔMokmt6* grown *under in vitro* complete medium (CM).**
(XLSX)

**S2 Table. Location and information of H3K27me3 ChIP peaks called from MACS2.**
(XLSX)

**S3 Table. Location and information of H3K27ac ChIP peaks called from MACS2.**
(XLSX)

**S4 Table. Gene Ontology for H3K27me3 marked genes identified for *M. oryzae* strain Guy11 grown *in vitro* complete medium.**
(XLSX)

**S5 Table. Kyoto Encyclopedia of Genes and Genomes (KEGG) analysis of H3K27me3 marked genes.**
(XLSX)

**S6 Table. Primers used in this study.**
(XLSX)

**S7 Table. Contingency table for chi-square test of association between predicted effectors and H3K27me3 domains.**
(XLSX)

**S8 Table. Contingency table for chi-square test of association between predicted effectors induced during infection and H3K27me3 domains during complete medium growth.**
(XLSX)

**S9 Table. High-confidence characterized effectors, avirulence and *in planta* induced genes with H3K27me3 status.**
(DOCX)

**S10 Table. Contingency table for chi-square test of association between high-confidence characterized effectors and H3K27me3 domains.**
(XLSX)

**S11 Table. Contingency table for chi-square test of association between high-confidence characterized effectors and transcriptional induction in △*MoKMT6*.**
(XLSX)

## Acknowledgments

We would like to thank Dr. Barbara Valent and Mrs. Melinda Dalby (Kansas State University) for their discussion of the work, sharing fungal strains and helpful resources and protocols. We thank Dr. Alina Akhunova and Mrs. Jie Ren for their support through the Integrated Genomics Facility (Kansas State University). Thanks to Dr. Zonghua Wang and Dr. Jiandong Bao (Fujian Agriculture and Forestry University) for providing an early version of the Guy11 genome assembly. Thanks to Dr. Shahideh Nouri (Kansas State University) and their lab members (Dr. Maryam Rastegar and Mrs. Carla Redila) for generous sharing of CFX 96 Real-Time thermocycler.

## Author Contributions

**Conceptualization:** Wei Zhang, Jun Huang, David E. Cook.

**Data curation:** Wei Zhang.

**Formal analysis:** Wei Zhang, Jun Huang, David E. Cook.

**Funding acquisition:** David E. Cook.

**Methodology:** Wei Zhang, Jun Huang, David E. Cook.

**Visualization:** Wei Zhang, Jun Huang.

**Writing – original draft:** Wei Zhang, Jun Huang, David E. Cook.

**Writing – review & editing:** Wei Zhang, Jun Huang, David E. Cook.

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
