## [Decision Letter · Decision Letter 0]

16 Oct 2020

Dear Dr Cook,

Thank you very much for submitting your Research Article entitled 'Histone modification dynamics at H3K27 result in altered transcription of in planta induced genes in Magnaporthe oryzae' to PLOS Genetics. Your manuscript was fully evaluated at the editorial level and by independent peer reviewers. The reviewers appreciated the attention to an important problem, but raised some substantial concerns about the current manuscript. Based on the reviews, we will not be able to accept this version of the manuscript, but we would be willing to review again a much-revised version. We cannot, of course, promise publication at that time.

As you will see from the detailed comments of the reviewers, they all like the approach taken, appreciate the quality of the presented data and find the manuscript informative. However, they have a long list of questions and indicate many aspects where the work and the data presentation can be improved, and the conclusions strengthened. The questions address several methodological details, the specificity of the mutant effects, and naturally the limitations of analysis after infection in planta. The latter is an inherent constraint, but there are some suggestions how to get closer at least for some target genes. All reviewers bring up the point that there are so few effector candidates investigated, although many more are expected to exist, and whether the H3K27me3 regulation is (or is not) relevant for others. It is also asked how much the pathogenicity of the fungus is affected by the histone modifier mutations, and the reviewers suggest a closer look into the functionality of the affected genes.

If you decide to revise the manuscript for further consideration at PLOS Genetics, please aim to resubmit within the next 60 days, unless it will take extra time to address the concerns of the reviewers, in which case we would appreciate an expected resubmission date by email to plosgenetics@plos.org.

[LINK]

We are sorry that we cannot be more positive about your manuscript at this stage. Please do not hesitate to contact us if you have any concerns or questions.

Yours sincerely,

Ortrun Mittelsten Scheid

Associate Editor

PLOS Genetics

John Greally

Section Editor: Epigenetics

PLOS Genetics

Reviewer's Responses to Questions

**Comments to the Authors:**

Reviewer #1: In this study, Zhang et al. describe genome wide histone modification dynamics (H3K27me3, H3K27ac and H3K36me3) under in vitro growth conditions in the plant pathogenic fungus Magnaporthe oryzae. The authors generate deletion mutants of several PRC2 components and the acetyltransferase Gcn5 to analyze genome wide H3K27 dynamics and their transcriptional implications in vitro and in planta. They find that transcriptional repression in vitro correlates to the presence of H3K27me3 and lack of H3K27ac. Loss of H3K27me3 in the kmt6 mutant results in de-repression and enrichment of H3K27ac in at least some of the regions. Among those de-repressed genes are known effectors and in planta induced genes. The manuscript has a clear structure and is well written.

Since in-planta ChIP was not performed, there is no proof of actual chromatin remodeling in planta but only a correlation between in vitro chromatin states and in planta transcription. This correlation has been shown before in other plant pathogens. Without additional ChIP-seq experiments in planta or at least comparable conditions, the findings in this manuscript do not provide significant advances over previous correlations of in planta induced genes and in vitro chromatin states.

Comments:

The PRC2 mutants show an altered phenotype upon growth on RPA and to a lesser extent on CM. What was the reason to choose CM as growth medium for ChIP and RNA-seq and not RPA? What stage in the life or infection cycle of Magnaporthe comes close to growth in CM?

Since ChIP-seq in planta is challenging and to date there is no study that goes beyond correlation of ChIP-seq and RNA-seq data from different growth conditions, it would be great to show at least different in vitro conditions that result in chromatin and transcriptional changes. Is there any kind of growth media (minimal medium, containing rice leaves etc.) that alters the transcriptional program and would allow to detect chromatin changes? Although this would of course not be identical to a real plant infection, this could be an opportunity to detect local chromatin changes that reflect similar changes during infection.

Removal of all H3K27me3 like in the kmt6 mutant is an highly artificial chromatin state that likely has many effects on expression that are not comparable to local chromatin remodeling during plant infection.

Does deletion of kmt6 or gcn5 affect pathogenicity on the plant?

Are the differences in signal intensity of H3K27me3 on TEs correlated to copy number or mapping of reads to TEs?

In Fig 3B it appears that genes that are normally marked by H3K27me3 in WT, gain H3K27ac in the kmt6 mutant. Conversely, ~ 10 % of non-marked genes lose H3K27ac signal. What kind of genes lost H3K27ac in the kmt6 mutant and does this correlate to the expression profile?

How was the percentage of marked genes or TEs determined? Overlap or proximity of peaks and genes or promoters?

Are specific functional gene categories enriched among H3K27me3 marked genes?

Is there a difference between Figure 3C and S7A?

Since Gcn5 is not only responsible for acetylation of H3K27 but many other lysines on different histone tails and also interacts with RNA polymerase II, it is very difficult to determine if presence/absence of H3K27ac has any functional consequences. There may very well be a correlation to the transcriptional state of the locus but there is little evidence that H3K27ac alone has functional implications here.

To better understand dynamics of Gcn5 and H3K27me3, a single deletion mutant of Gcn5 and ChIP-seq profiles of H3K27me3 would be a relevant addition.

Are the 27 (presumed) effectors all known or predicted effectors in Magnaporthe? If there are more predicted effector, how many are marked by H3K27me3?

Figure 6B: Since there is no in planta ChIP-seq data available, ChIP qPCR of candidate genes in planta would give an idea about changing chromatin states here.

Line 284: I don’t think it is correct to compare activation rate in a very small selected subset of genes to the genome-wide number of upregulated genes in the kmt6 mutant.

Reviewer #2: In this work, Zhang and colleagues describe histone modification dynamics at H3K27 site regulate gene transcriptional landscape. In particular, those in planta induced genes are subjected to H3K27me3/H3K27ac regulation in Magnaporthe oryzae. The authors showed that H3K27me3 provides significant local transcriptional repression and contributes to normal in vitro growth by deleting and complementing the H3K27me3 methyltransferase complex PRC2 core components. Moreover, H3K27ac was redistributed to previously occupied H3K27me3 sites, was required for transcriptional activation of roughly a third of differentially expressed genes. The authors also observed that many in planta induced genes were marked by H3K27me3, and detailed how altered H3K27 modification influences transcription of in planta induced genes. The data is carefully analyzed and presented, and I am fine with their statement of the major conclusion. Overall, this is an interesting paper that merit the publication. However, it seems to me that the story still has holes and I hope they could address these issues.

Major concerns:

(1) Regarding that M. oryzae is an important rice pathogen and gene expression pattern of in planta induced genes are impaired in mutants. The paper didn’t show infection results, what about the pathogenicity of different M. oryzae mutants that are used in this research?

(2) The authors found histone modifications at H3K27 contribute to regulation of genes important during host infection. For example, many effector genes were up-regulated in H3K27me3 deficient mutants. What about the H3K27me3 and H3K27ac level of these induced genes during infection in WT and mutants? These important data are missing in the current manuscript. Without these data, it is hard to set up a link between gene expression and the modification changes. I suggest authors to show a few cases by adding a few experimental data.

other concerns:

(1) The authors observed that H3K27ac was negatively correlated with H3K27me3(r=-0.61, p < 2.2e-16) and H3K36me3 was positively correlated with H3K27me3(r= 0.61, p < 2.2e-16). H3K36me is well known as a activate mark, is H3K36me3 a repressive mark in M. oryzae? What about the expression change of genes that lost H3K36me3 in Mokmt6 mutant?

(2) Do some genomic regions gain H3K27ac but lost H3K36me3 at the same time? What about these genes expression change compare to genes gained H3K27ac or lost H3K36me3 only?

(3) GCN5 can acetylate H3K9ac, H3K14ac, H3K18ac, H3K23ac, H3K27ac and so on, the authors found many H3K27me3 marked genes activated by H3K27ac, how to exclude crosstalk of H3K27me3 and other histone acetylation?

(4) In fig6, the authors analyzed 27 effector gene that are presented in M. oryzae strain Guy11. What about other secreted proteins? I assumed that M. oryzae should have many effector genes.

(5) Many in planta induced genes were up-regulated in ΔMokmt6, what about the expression change of these genes in ΔMokmt6 during infection?

Reviewer #3: Understanding how genes from pathogenic and endophytic fungi are specifically upregulated/derepressed in planta remains an important question to be answered. But what is clear from work published to date is the important role specific histone marks play in controlling chromatin structure and gene expression in planta. An important limitation to addressing this question is the technical difficulty in studying histone modification dynamics at a global scale during plant infection as noted by the authors in the Discussion of this manuscript. Working within these constraints the authors report on some important new insights into the role of chromatin structure changes in the plant pathogen Magnaporthe oryzae by specifically focusing on the dynamics of histone H3K27 modification. While much of the work examines changes that occur ex planta, which uncovers good information about the repressive state of plant induced genes, the authors are still able to provide some new insights into what might be happening in the plant by aligning global changes in histone dynamics in axenic culture with in planta transcriptomics using wild type and mutants defective in H3K27 methylation or acetylation. The work is of a high technical standard and a very important contribution to the growing literature in this field.

ChIP analysis of H3K27me3 and H3K27ac signals across the genome (Fig 1) shows that these two marks have distinct distribution profiles with a predominance of the former in regions of the genome enriched with TE of the Class I type. The H3K27ac were almost exclusively associated with genes actively transcribed in axenic culture. These experiments provide important genome scale insights into the distribution of these two histone marks and highlight the importance of H3K27me3 as a repressive mark.

The authors then go onto show by deletion analysis that three components of the PRC2 complex are required for H3K27 trimethylation with a statistically significant effect on vegetative growth (Fig 2).

Using the kmt6 deletion mutant they go on to show that loss of H3K27me3 results in a specific redistribution of H3K27ac marks (Fig 3). However, as highlighted in the Discussion the mechanism underlying this redistribution is not clear. Interestingly, they found that H3K36 methylation marks correlate with H3K27me3 marks, and these are lost in the kmt6 mutant, providing good evidence for cross talk between these two methylation modifications. The text in this section (around line 159) could be clearer by specifically mentioning whether they are looking at changes in marks in WT or mutant. The subsequent cluster analysis used to examine more deeply the chromatin dynamics provides further support for reciprocity between H3K27me3 and H3K27ac and cross talk with H3K36me3. However, the difference in colour coding between the upper and lower panels of Fig 3C does make it difficult for the reader to interpret this Figure – there is either an error here or I have not understood the difference?

RNAseq is then used to examine transcriptome differences between WT and the kmt6 mutant and between WT and the kmt6/gcn5 double mutant. The high percentage of the upregulated genes (87.9%) in the mutant that have H3K27me3 marks provides strong evidence that the Kmt6 H3K27 methylase is required for the H3K27 methylation observed and gene repression. However, the evidence that acetylation is required for expression at these gene loci was not so strong given only a third of the K27me3 marked genes that were upregulated in the kmt6 mutant were downregulated in the kmt6/gcn5 double mutant. I think it is important for the culture phenotype of the double mutant to be reported as I suspect the double mutant will impact on growth and development. Also, a little more information on the specificity of the gcn5 gene product is needed. I note in the discussion it is described as a general acetyl transferase but in the results section it is said to be the major histone acetyl transferase for H3K27 acetylation. The integration of the RNAseq data with ChIP-seq data provides further support for the connection between the H3K27 modification status and gene expression – very nice work.

The final two sections of the Results attempt to bridge the gap between what is observed in axenic culture and what might be happening during plant infection. Having identified the H3K27 marked genes in the earlier experiments the authors next examine how the expression of these genes changes in planta. Of the ~1000 genes differentially expressed in planta compared to axenic culture 44% of these fall within the H3K27 marked genes further highlighting the importance of this mark in controlling gene expression between the two physiological conditions. Two thirds of these H3K27 marked genes were up-regulated in the kmt6 mutant. Taken together this section highlights the importance of this mark for repression of this subset of genes in culture and the need for removal of this mark for high level of expression in planta. In presenting this work I would discourage the authors from talking about genes being up-regulated in planta when the comparison is with expression in culture which is a totally different and discontinuous physiological condition. I think it is acceptable to talk about up-regulation when comparing gene expression in culture for mutant and WT.

Finally, the authors hone in on the genes encoding known and putative effectors in M. oryzae. I was surprised that there are just 27 known or presumed effectors present in M. oryzae? This seems a very small number compared to the numbers identified in other fungal plant pathogens. While this analysis does highlight the potential importance of H3K27 modifications in regulating the chromatin state and expression of some effector genes in planta the regulation of others is clearly more complex. This is perhaps not surprising given this group of genes will be required at different stages of infection during colonization of the host. I thought the hierarchical clustering approach used for this analysis was good. Although the results are complex they do highlight the important role H3K27methylation plays in controlling expression of plant-induced genes. One important limitation of comparing gene expression in culture with that in planta and drawing conclusions about the chromatin state that should be highlighted is the growth conditions used ex planta are invariably unlike what will be found in planta. This is an important physiological limitation than many papers fail to mention.

In summary, this is a comprehensive piece of work carried out to a high technical standard that provides important new insights into the role of histone modification dynamics in controlling gene expression in the plant pathogen M. oryzae. As discussed above there are some major technical challenges in examining these dynamics in planta but despite this limitation this work increases our understanding of the importance of the H3K27 modifications in controlling gene expression.

**Have all data underlying the figures and results presented in the manuscript been provided?**

Reviewer #1: Yes

Reviewer #2: Yes

Reviewer #3: Yes

PLOS authors have the option to publish the peer review history of their article (what does this mean?). If published, this will include your full peer review and any attached files.

Reviewer #1: No

Reviewer #2: No

Reviewer #3: No

---

## [Decision Letter · Decision Letter 1]

22 Jan 2021

Dear Dr Cook,

We are pleased to inform you that your manuscript entitled "Histone modification dynamics at H3K27 are associated with altered transcription of in planta induced genes in Magnaporthe oryzae" has been editorially accepted for publication in PLOS Genetics. Congratulations!

Yours sincerely,

Ortrun Mittelsten Scheid

Associate Editor

PLOS Genetics

John Greally

Section Editor: Epigenetics

PLOS Genetics

Comments from the reviewers (if applicable):

Reviewer's Responses to Questions

**Comments to the Authors:**

Reviewer #1: All of my comments have been addressed by the authors.

Reviewer #3: The revision is a significant improvement with all the issues I raised as reviewer 3 addressed. I am pleased to see the additional data added with Figs S13 and S14, Fig 3 modified and text edited for clarification.

**Have all data underlying the figures and results presented in the manuscript been provided?**

Reviewer #1: Yes

Reviewer #3: Yes

PLOS authors have the option to publish the peer review history of their article (what does this mean?). If published, this will include your full peer review and any attached files.

Reviewer #1: No

Reviewer #3: **Yes: **Barry Scott

**Data Deposition**

http://datadryad.org/submit?journalID=pgenetics&manu=PGENETICS-D-20-01354R1

**Press Queries**

---

## [Editor Report · Acceptance letter]

29 Jan 2021

PGENETICS-D-20-01354R1 

Histone modification dynamics at H3K27 are associated with altered transcription of in planta induced genes in Magnaporthe oryzae 

Dear Dr Cook, 

We are pleased to inform you that your manuscript entitled "Histone modification dynamics at H3K27 are associated with altered transcription of in planta induced genes in Magnaporthe oryzae" has been formally accepted for publication in PLOS Genetics! Your manuscript is now with our production department and you will be notified of the publication date in due course.

With kind regards,

Alice Ellingham

PLOS Genetics

On behalf of:
